# Snake venom-defined fibrin architecture dictates fibroblast survival and differentiation

Zhao Wang [1], Jan Lauko [1], Amanda W. Kijas[1], Elliot P. Gilbert [1,2], Petri Turunen[3], Ramanathan Yegappan [1], Dongxiu Zou[1], Jitendra Mata [2] & Alan E. Rowan[1] ✉

Fibrin is the provisional matrix formed after injury, setting the trajectory for the subsequent stages of wound healing. It is commonly used as a wound sealant and a natural hydrogel for three-dimensional (3D) biophysical studies. However, the traditional thrombin-driven fibrin systems are poorly controlled. Therefore, the precise roles of fibrin's biophysical properties on fibroblast functions, which underlie healing outcomes, are unknown. Here, we establish a snake venom-controlled fibrin system with precisely and independently tuned architectural and mechanical properties. Employing this defined system, we show that fibrin architecture influences fibroblast survival, spreading phenotype, and differentiation. A fine fibrin architecture is a key prerequisite for fibroblast differentiation, while a coarse architecture induces cell loss and disengages fibroblast's sensitivity towards TGF-β1. Our results demonstrate that snake venom-controlled fibrin can precisely control fibroblast differentiation. Applying these biophysical principles to fibrin sealants has translational significance in regenerative medicine and tissue engineering.

Fibrin is the provisional matrix formed after injury, which is abundant in plasma and plays a primary role in the early stage of wound healing. The broader applications of fibrin hydrogels in tissue engineering and biophysical studies have been hampered due to batch-to-batch variations, limited control of structural and mechanical properties, and rapid degradation[1–3]. Although it is now well established that the biophysical cues of the extracellular matrix (ECM), including topology, bulk stiffness, stress relaxation, stress stiffening, and matrix architecture, all play a role in cellular behaviors, the influence of ECM three-dimensional (3D) architecture on cell behavior has not yet been fully elucidated[4–7]. This is due to the inability to precisely control the 3D architecture of ECM while decoupling the mechanical properties under physiological conditions[8–12].

The provisional fibrin matrix, which can last up to 7 days in the wound sites, provides key biophysical cues to the early events such as fibroblast differentiation[13]. Fibroblast-to-myofibroblast differentiation plays a key role in wound closure and scarring[13]. The current theory is that both biochemical factors, such as transforming growth factor β1 (TGF-β1), and mechanical forces are prerequisites for myofibroblast activation[14–16]. However, these in vitro studies are mostly based on 2D cell culture models; how the biophysical cues of 3D matrices, especially the fibrin matrix, influence fibroblast functions and hence scarring are still elusive[17].

Persistent activation and prolonged survival of myofibroblasts have been seen as the hallmarks of hypertrophic scars and tissue fibrosis[18]. Emerging evidence has shown the therapeutic promise of fibroblast-targeted therapies in achieving regenerative wound healing[19]. For example, inhibiting the focal adhesion kinase (FAK) signaling of fibroblasts reduces scar formation in mice[20]. More recently, scarless wound healing has been achieved in mice models by either

[1]Australian Institute for Bioengineering and Nanotechnology, The University of Queensland, St Lucia, QLD 4072, Australia. [2]Australian Centre for Neutron Scattering, Australian Nuclear Science and Technology Organisation, Lucas Heights, NSW 2234, Australia. [3]Microscopy Core Facility, Institute of Molecular Biology, Mainz 55128, Germany. ✉e-mail: alan.rowan@uq.edu.au

activating adaptive immune responses or off-loading fibroblast mechanotransduction by targeting the Yes-associated protein (YAP) pathway or TGF-β1[21–23]. These pre-clinical studies implicate the important role of mechanotransduction in modulating fibroblasts and their functional roles, opening an avenue to designing biomaterials with specific biophysical parameters, which can promote scarless or regenerative wound healing.

We have developed a snake venom-controlled 3D fibrin system with precisely controlled architectural and mechanical properties. Employing the snake venom-derived protein ecarin, which rapidly activates the thrombin precursor prothrombin[24], we established a tunable fibrin network system that can access a broader range of pore sizes, compared to the existing thrombin-initiated system in which fibrin was initiated by thrombin directly[25]. Control over the cell-adhesive ligand density in fibrin was achieved by titration of purified fibronectin. By tuning the crosslinks within the fibrin fibers using the coagulation factor XIII (FXIII), the mechanical properties can be fine-tuned. Furthermore, the implementation of a second snake venom-derived protein, textilinin, which inhibits fibrinolysis-induced fibrin degradation[26] ensured the stability of the prepared networks.

Employing this defined system, we find that a coarse fibrin architecture promotes early fibroblast spreading but induces cell loss. In contrast, a fine architecture renders fibroblast highly sensitive to TGF-β1, triggering profound fibroblast-to-myofibroblast differentiation in matrix stiffness and cell contractility-dependent manners. These findings implicate fibrin architecture as a key determinant of fibroblast functions and provide biophysical strategies for designing biomaterials to treat fibrosis and promote scarless wound healing.

## Results

### Precisely tuned fibrin 3D architecture by the snake venom protein ecarin

Current methods for fibrin network formation rely on the direct activation of fibrinogen by thrombin. This approach mimics the last step of the in vivo blood coagulation pathway (Fig. 1a in green). By changing the fibrin polymerization kinetics via tuning the thrombin activity, the fibrin architecture can be modulated[27,28]. However, the main limitation of this approach lies in the difficulty of forming a coarse network (pore sizes >4 μm) under physiological conditions in a relatively short time (<1 hour) without significant salt, pH, or other formulation changes[25,28–30]. Using purified fibrinogen and initiating fibrin networks with varying concentrations of thrombin under physiological conditions, we observed that a decreased concentration of thrombin (from $10^{-2}$ to $10^{-4}$ Unit μL$^{-1}$) resulted in only marginally more coarse networks (Fig. 1b, left column), while the gelation lag time (time needed to form the initial network after initiation) increased from ~15 s to ~13 min (Supplementary Fig. 1). No network was detected when the thrombin concentration was decreased to $10^{-5}$ Unit μL$^{-1}$ (Fig. 1b). By implementing a delay of only 5 min before the network initiation (thrombin pre-heated for 10 min at 37 °C), it was observed that thrombin failed to form any network even at a concentration of $2 \times 10^{-4}$ Unit μL$^{-1}$ (Fig. 1b, right column), suggesting that this traditional thrombin-initiated fibrin system is highly enzyme-activity sensitive, unstable, and limited over diverse network architectures.

To overcome these limitations, we designed a system that closely mimics the in vivo coagulation pathway. The fibrinogen activation was initiated one step earlier by replacing thrombin with its precursor, prothrombin, and a recombinant snake venom-derived protein, ecarin[24] (Fig. 1a in yellow). Ecarin directly activates prothrombin without any co-factors, replacing the uncontrollable activation step of the prothrombinase complex (Fig. 1a in grey). Consequently, the thrombin cleavage activity is continuously replenished from the pro-thrombin pool. As shown in Fig. 1c, defined fibrin networks with tunable architecture were obtained by only tuning the prothrombin and/or ecarin concentrations. Nine conditions were chosen (Sample 1–9,

hereafter referred to as S1–9) and fully characterized. The architectural features of these networks (S1–9) were quantified by both confocal laser scanning microscopy (CLSM) and combined small-angle neutron scattering/ultra-small angle neutron scattering (SANS/USANS) techniques[31]. The latter quantifies the hierarchical structures of the fibrin network across multiple length scales ranging from protofibrils to the overall network (Supplementary Fig. 2a). Both analyses reveal that higher concentrations of prothrombin and/or ecarin result in denser networks with smaller pore sizes (Fig. 1d, from ~13 to ~1 μm) and thinner fibers (Fig. 1e, radius from ~650 to ~100 nm). Additional ultra-structural properties including the fiber density (Supplementary Fig. 2b), the surface area of the fibers (Supplementary Fig. 2c), the number of protofibrils per fiber (Supplementary Fig. 2d), and the network fractal dimension (Supplementary Fig. 2e), were also evaluated from the SANS/USANS data and compared with the thrombin-initiated system. These data highlight that the snake venom-controlled fibrin system can be precisely tuned and has a much broader range of accessible network pore sizes, fiber diameters, and surface areas than the conventional thrombin-initiated system.

To further understand the mechanisms that lead to the differences between the two systems, we investigated the dynamic cleavage activity of thrombin for the thrombin-initiated system or newly formed thrombin by ecarin using a chromogenic reporter activity assay. As shown in Supplementary Fig. 3, the absorbance curve $A_{(405-490nm)}$ of the snake venom-controlled system exhibited an S-shaped profile (sigmoidal), with the cleavage rate $\Delta A/min$ increasing over more than 60 min for the S6 network condition. By contrast, the absorbance curve $A_{(405-490nm)}$ of the thrombin-initiated system showed an increasing trend with decreasing cleavage rate $\Delta A/min$, which only persisted for 20 min when at $10^{-2}$ Unit μL$^{-1}$. Based on this activity assay, we estimated a threshold for the absorbance reading ($A_{(405-490nm)}$ ~0.035), below which no networks were formed (Supplementary Fig. 3b). The time point when $A_{(405-490nm)}$ reached the threshold (horizontal lines in Supplementary Fig. 3b) was comparable to the gelation lag time (Supplementary Fig. 1). These observations demonstrate that the characteristic S-shaped kinetic profile of the snake venom-controlled system enables access to a coarser network by generating a lower cleavage activity with a longer gelation lag time compared to the thrombin-initiated system (shown in Fig. 1f).

### Precisely tuned mechanical properties of fibrin networks

To investigate the long-term mechanical properties of the fibrin networks, we depleted the fibrinolysis-induced degradation by employing a recombinant snake venom-derived protein textilinin[26], a potent inhibitor of the fibrinolytic pathway (Fig. 1a in the red rectangle). We then monitored the long-term (≥6 hours) mechanical properties of the established fibrin networks (S1–S9) by shear-stress rheology. In this time-resolved experiment, we monitored dynamic storage modulus G′ (Fig. 2a), matrix stiffness (endpoint G′ at 6 hours, Fig. 2b), and the viscoelastic property indicated by loss factor tanδ (calculated as endpoint G″/G′ at 6 hours, Fig. 2c). As the prothrombin concentration increased from $10^{-7}$ to $10^{-4}$ IU μL$^{-1}$ (S1, S2, S3, and S5), the matrix stiffness and loss factor remained comparable at ~400 Pa and ~0.02–0.03, respectively (Fig. 2a–c). Interestingly, when the pro-thrombin concentration was increased from $10^{-4}$ to $10^{-3}$ IU μL$^{-1}$ (S5 vs. S8), the matrix stiffness was almost halved while the loss factor was doubled; by tuning the concentration of ecarin from 0.01 nM to 3 nM, a similar trend in matrix stiffness and loss factor was seen, highlighting that for the same concentration of fibrin, the mechanical properties of the networks S1–S6 are comparable; the extremely fine networks S8 and S10 (an extra condition initiated by $10^{-4}$ IU μL$^{-1}$ of prothrombin and 3 nM of ecarin) are intrinsically softer and more viscous than the coarse networks, which is similar to other studies[29,32].

The transglutaminase factor XIII (FXIII) is known to form covalent crosslinks between protofibrils within fibrin fibers, leading to the

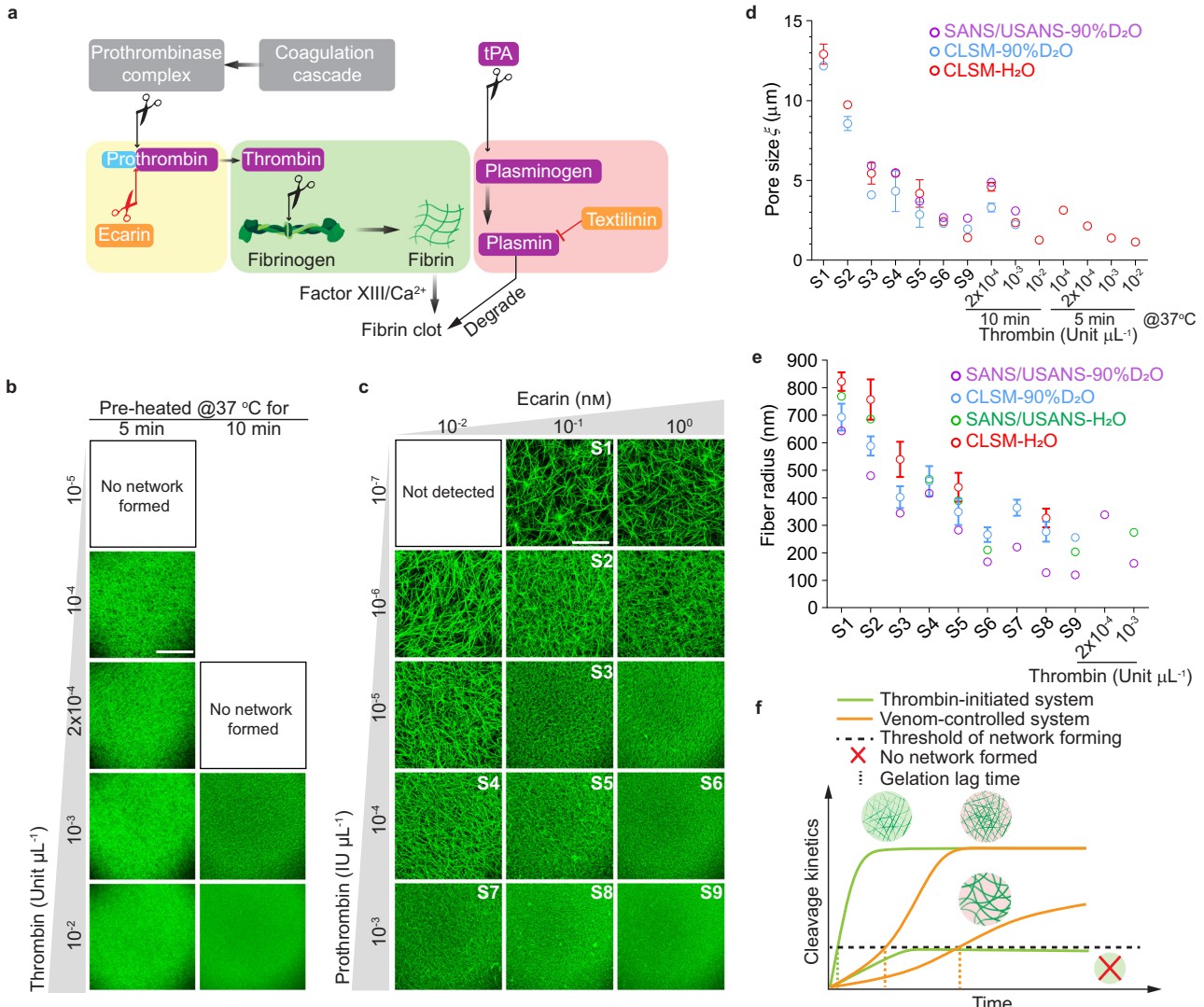

**Fig. 1 | Architectural comparisons between the snake venom-controlled and thrombin-initiated fibrin systems. a** Illustration of the blood coagulation pathway and the fibrinolytic pathway in wound healing. The snake venom-controlled (in yellow and red rectangles) system and the thrombin-initiated system (in the green rectangle) are highlighted. tPA, tissue plasminogen activator. **b** Confocal laser scanning microscopic (CLSM) images of the fibrin networks initiated by varying concentrations of thrombin after being pre-heated at 37 °C for 5 min (left column) or 10 min (right column). **c** CLSM images of the fibrin networks formed by the ecarin/prothrombin system after being pre-heated at 37 °C for 5 min. For each image in **b** and **c**, a maximum image projection of a 290 μm × 290 μm area with 40 μm thickness (1 μm/frame) of the fibrin network (4 mg mL$^{-1}$) is shown. Scale bar, 100 μm. Architectural properties, including pore size (**d**) and fiber radius (**e**) of the

snake venom-controlled and thrombin-initiated fibrin networks characterized by CLSM analysis and combined SANS/USANS measurements (purple circles), are shown. For pore size analysis, fibrin networks formed in different aqueous buffers, including H$_2$O (red circles) and 90% D$_2$O (blue circles), are compared. For fiber radius analysis, fibrin networks formed in H$_2$O (green circles) are also compared. CLSM data in **d**–**e** are shown as mean value ± SD from 3 biologically independent experiments ($n = 3$). SANS/USANS data in **d**–**e** are from 1 experiment ($n = 1$). **f** Illustration of the cleavage kinetics and the resulting fibrin networks by the snake venom-controlled (in yellow) and the thrombin-initiated (in green) systems. In the snake venom-controlled system, a coarse network is formed and characterized as a network with large pore sizes and thick fibers, while a fine network has more branched and thin fibers.

long-lasting stiffening of the fibrin clots after network formation[25]. We utilized FXIII as an additional tool to modulate the mechanical properties of the fibrin networks by addition of plasma-derived purified FXIII (pFXIII) or its synthetic inhibitor D004, a transglutaminase inhibitor that acetylates the active site cysteine[33]. We found that the inhibition of crosslinks by D004 (0–10 μM) resulted in softer fibrin networks (Supplementary Fig. 4a, matrix stiffness decreased from ~400 to ~200 Pa) with a higher loss factor (Supplementary Fig. 4b), while the enhancement of crosslinks by adding FXIII induced the opposite (matrix stiffness increased from ~400 to ~1000 Pa). Moreover, the addition of the inhibitor D004 (≤10 μM) did not affect the pore size of the networks, whilst the addition of pFXIII (60 μg mL$^{-1}$) induced denser networks with smaller pore sizes (Supplementary Fig. 4c). This allows

us to decouple the stiffness and pore size of the fibrin networks simply by tuning the D004 (≤10 μM) or pFXIII (≤ 20 μg mL$^{-1}$) concentration.

## Achieving biophysical tunability and reproducibility of fibrin networks

The above data was detected using one batch of fibrinogen purified from pooled human plasma (B4a in Supplementary Fig. 5 and Supplementary Table 1). Significant batch-to-batch variation in matrix architecture was observed when analyzing additional batches of fibrinogen, despite a standardized fibrinogen purification and fibrin formation protocol employed (see Methods and concluded in Supplementary Fig. 19a). Studying in more detail, we identified that the endogenous level of FXIII varied among our purified fibrinogen

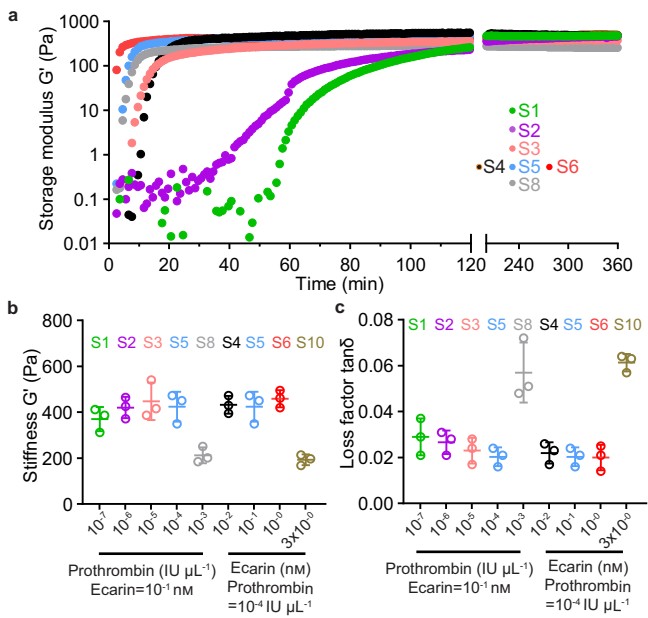

**Fig. 2 | Mechanical properties of the snake venom-controlled fibrin networks.**
**a** Dynamic storage modulus G′ of the snake venom-controlled fibrin networks (S1–S6 and S8) were monitored over 6 hours after initiation. Matrix stiffness characterized by the endpoint storage modulus G′ (**b**) and loss factor tanδ (**c**) of different networks were determined after the mechanical properties of the fibrin networks were stable 6 hours after initiation. Data in **a** are from 1 representative experiment. Data in **b**–**c** are shown as mean value ± SD from 3 independent measurements ($n = 3$).

batches (Supplementary Fig. 5 and Supplementary Table 1) and commercial fibrinogen products, as reported elsewhere[34]. Moreover, the variation of endogenous FXIII activity in different batches of fibrinogen was related to both the architectural (Fig. 3a for pore size) and mechanical (Fig. 3b for stiffness and Fig. 3c for loss factor) variations. However, the FXIII activity did not overtly influence the gelation lag time (circles in Supplementary Fig. 6). These observations highlight that the endogenous FXIII activity in fibrinogen contributes to the biophysical batch-to-batch variations of the formed fibrin networks.

To demonstrate a standardized approach, we tuned the FXIII activities of three batches of fibrinogen (with either low (B7a), medium (B7c), or high (B8b) endogenous FXIII activities with batch information shown in Supplementary Table 1) by the addition of pFXIII or its inhibitor D004. The architectural and mechanical properties of the formed networks for conditions S4 and S6 were evaluated and compared with the untuned values in Fig. 3a–c. We found that the tuned pore size (Fig. 3d), matrix stiffness (Fig. 3e), and loss factor (Fig. 3f) almost overlapped with the trendlines of untuned values, with the tuned gelation lag time showing no significant changes (dots in Supplementary Fig. 6). These results suggest that the standardization of FXIII levels is critical to achieving the tunability and reproducibility of biophysical properties between fibrinogen batches.

Fibronectin, an additional significant glycoprotein in blood and incorporated in fibrin during clotting, provides abundant ligands for cell attachment and plays a vital role in wound healing[35]. Its concentration varies in different fibrinogen products and human plasma (normal range of 0.2–0.6 mg mL⁻¹)[36,37]. We, therefore, precisely tuned the levels of fibronectin by supplementing our purified fibrin system with purified fibronectin-fibrinogen complex (pFn-pFg, hereafter referred to as pFn, details in Supplementary Fig. 7), which enables the concentration of fibronectin in the fibrin network to be precisely tuned (Supplementary Fig. 7a). The amount of fibronectin incorporated into fibrin networks was evaluated by estimating the proportion in the gel

component (Supplementary Fig. 7b). We estimated that ~51–62% of the fibronectin was incorporated in the fibrin networks examined (Supplementary Fig. 7b). The fibronectin component and its distribution in fibrin networks were also detected by immunostaining to examine its participation in these hybrid networks (Supplementary Fig. 7c). We found that ~70–80% of fibronectin fluorescence intensity was colocalized with fibrinogen (Supplementary Fig. 7d). These results suggest that fibronectin is incorporated in the fibrin fibers, as previously reported[38]. Furthermore, we observed that a low concentration of fibronectin (≤0.4 mg mL⁻¹ at a constant fibrinogen concentration of 4 mg mL⁻¹) did not overtly change the pore size or the bulk stiffness of the formed fibrin networks (S4 and S6, Supplementary Fig. 7e, f). This allowed the ligand density of the fibrin network to be independently tuned by adding fibronectin (≤0.4 mg mL⁻¹) without substantially changing other biophysical cues.

Matrix degradation-mediated stress relaxation is another vital biophysical cue that regulates cell behaviors[39,40]. We evaluated the degradation of S4 and S6 networks formed as a function of different levels of FXIII activities (tuned by pFXIII or D004). We found that under all conditions, S4 networks had a longer degradation half-time than the corresponding S6 networks, suggesting that a fine architecture is more sensitive to degradation than a coarse network (Supplementary Fig. 8a, b). Moreover, the enhanced crosslinking imparted by the addition of pFXIII increased the resistance to degradation; conversely, reduced crosslinking after the D004 addition facilitated degradation in both S4 and S6 networks (Supplementary Fig. 8a, b). To control the degradation, we employed textilinin to inhibit the fibrinolytic pathway. By tuning the concentration of textilinin, the stability of the fibrin networks was maintained from several hours to >7 days, as demonstrated by turbidimetry measurements (Supplementary Fig. 8c). After optimization, we selected 10 μM textilinin to deplete fibrinolysis-induced degradation for long-term cell culture in this study.

Lastly, we evaluated the permeability and diffusivity of the 3D fibrin networks that effects the movement of soluble factors (such as signaling molecules and nutrients), employing approaches as previously described[41]. A permeation assay outlined in Supplementary Fig. 9a was used to determine the permeability coefficient K. We found that S4 fibrin networks had a higher permeability coefficient K of $4.7 \pm 0.9 \times 10^{-8}$ cm² than the smaller pore size of the S6 networks ($1.4 \pm 0.3 \times 10^{-8}$ cm²) under hydrostatic pressure (Supplementary Fig. 9b). By evaluating the intensity of permeabilized fluorescently labeled dextran (10 kDa and 70 kDa) in the eluate, we found that under hydrostatic pressure, both the low and high MW dextran flowed through fibrin networks largely unhindered, within 30–40 min with the rate being faster through the larger pore size of the S4 networks (Supplementary Fig. 9c). A second approach to examine the dextran diffusion through 3D fibrin networks without hydrostatic pressure was also employed, as outlined in Supplementary Fig. 9d. Within 4 hours, we observed that ~30% of the dextran had diffused through the fibrin and ~80% within 24 hours, the predicted maximum diffusion amount (Supplementary Fig. 9e), with no significant differences observed between S4 and S6 networks. These results demonstrate that although the permeability of S4 is higher than S6 networks due to the higher flow rates for the S4 network under hydrostatic pressure, both S4 and S6 fibrin networks can efficiently diffuse the nutrients/waste and biochemical factors with no substantial differences.

Together, these evaluations demonstrate and characterize a venom-controlled system for generating a defined 3D fibrin system for biophysical studies with high permeability, diffusivity and precisely and independently controlled pore size ($\xi$ = ~1–13 μm), matrix stiffness (~200–1000 Pa), ligand density (0–0.4 mg mL⁻¹ of incorporated fibronectin) and degradation time (from hours to days), while maintaining the fibrinogen at the same physiological concentration (4 mg mL⁻¹).

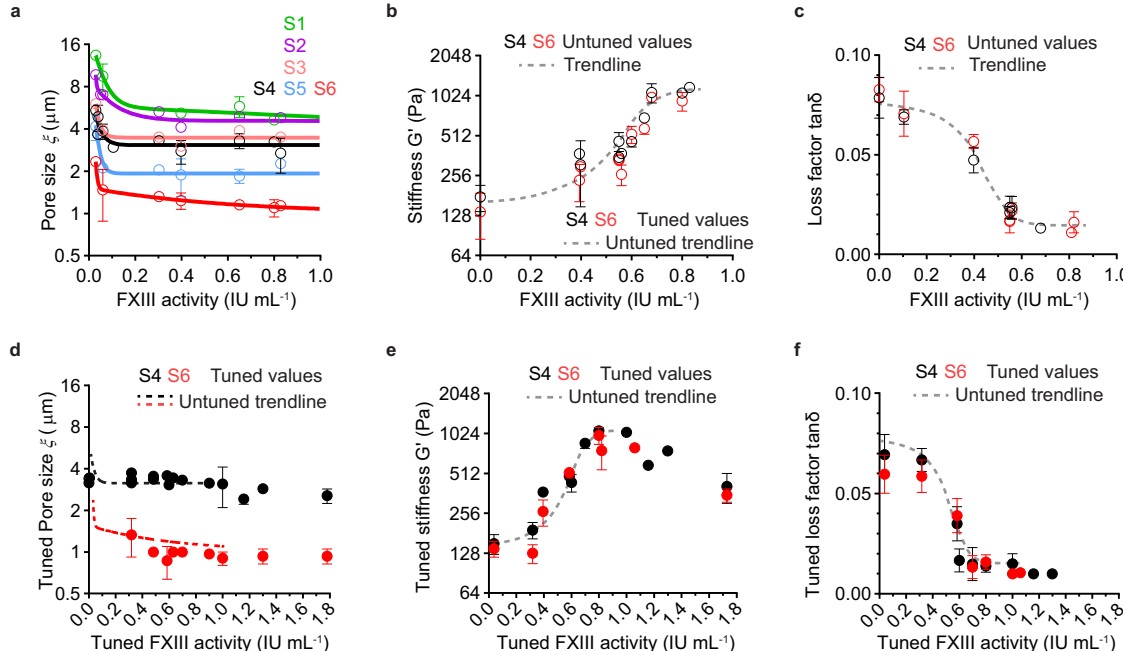

**Fig. 3 | Influence of FXIII activity and its tunability on architectural and mechanical properties of the fibrin networks.** Influence of endogenous FXIII activities of different batches of fibrinogen on the pore size (**a**), matrix stiffness (**b**), and loss factor (**c**) of the formed fibrin networks. Data are plotted as a function of the endogenous FXIII activity of each batch of fibrinogen, which is expressed as the FXIII activity per 1 mg mL$^{-1}$ of fibrinogen. Colored smoothed lines in **a** represent the trendlines of the untuned pore size values for individual networks (S1–S6). Grey dashed lines are the trendlines of untuned matrix stiffness (**b**) or loss factor (**c**) for the networks (S4 and S6) as a whole. Tuned properties of the fibrin networks (S4 and S6) were also detected after the FXIII activities were tuned by the addition of pFXIII or D004. Tuned pore size (**d**), matrix stiffness (**e**), and loss factor (**f**) were measured and plotted as a function of the tuned FXIII activities. The plotted tuned values are combined from three batches of fibrinogen with varied endogenous FXIII activities (B7a, B7c, and B8b shown in Supplementary Table 1). Dashed lines in **d**–**f** are the trendlines concluded in (**a**–**c**). Data in **a**–**f** for each condition are shown as mean value ± SD from 3 independent rheology measurements ($n = 3$).

## Fibrin 3D architecture dictates in vitro fibroblast survival and spreading

Previous studies have reported that fibroblast proliferation in 3D hydrogels (collagen and PEG) is much slower than in 2D culture and largely dependent on ligand density, pore size, and stiffness[32,42]. To investigate how the 3D biophysical cues of the fibrin matrix, especially architecture, influence fibroblast viability and growth, we employed our established defined fibrin system. We chose two distinctly different architectures (S4 as coarse networks with pore size ~3.3 μm, and S6 as fine networks with pore size ~1.1 μm) with independently controlled stiffness (ultra-stiff as ~1000 Pa, stiff as ~400–500 Pa, and soft as ~100–200 Pa, details shown in Supplementary Fig. 10) in the absence or presence of 0.4 mg mL$^{-1}$ fibronectin, which mimics the average value of fibronectin concentration in plasma. Fibroblasts were added to these networks ~5 min before the gelation time point according to the gelation lag time and 3D cultured for 7 days. Cellular responses, including cell viability, spreading morphology, and cell growth, were monitored by live/dead viability assays. We found that fibrin architecture significantly affected cell survival and spreading phenotype. We first compared the cell survival of the fibroblasts encapsulated in the fibrin networks without fibronectin. As shown in Fig. 4a and quantified in Fig. 4d, only 50–70% of the cells in the coarse networks survived in the first 24 hours, while no evident cells were lost in the fine networks. This architecture-induced cell death was only evident in the first 1–4 hours, as indicated by ethidium staining (Fig. 4a) and viability analysis (Fig. 4e). Surprisingly, the addition of fibronectin induced cell loss for the cells in the fine networks, while this influence was not evident in the coarse networks partly because the architecture-induced cell loss was more dominant in coarse networks. Fibroblast proliferation showed an increasing trend in the fine networks after day 3, while no cell growth was observed for the cells in the coarse networks (Fig. 4d). The cell viability was higher than 95% for the cells in

all the conditions after day 1, suggesting that the remaining cells were healthy (Fig. 4e). It was observed that the stiffness in 3D had a subtle influence on cell survival and proliferation, with no statistical difference in cell numbers between different matrix stiffnesses (Supplementary Fig. 11).

To quantify the cell morphology, we defined and measured cell length (x), cell width (y), and the elongation ratio (x/y) at different time points across different conditions (Fig. 4c). We found that cells spread quicker in coarse networks by comparing the cell length on early time points (from 1 hour to 4 hours, shown in Fig. 4a and quantified in Supplementary Fig. 12). After 5–7 days, cells in coarse networks exhibited a spindle-like elongated morphology, while cells in fine networks showed a more widespread phenotype, as depicted in Fig. 4c and quantified as increased cell length (Fig. 4f), decreased cell width (Fig. 4g), and higher elongation ratio (Fig. 4h). All these results suggest that though a coarse architecture promotes cell spreading, a fine architecture facilitates early cell survival and long-term cell growth.

Lastly, we evaluated the cytotoxicity of the two recombinant snake venom proteins, ecarin, and textilinin on various cell types of the skin to demonstrate broader biocompatibility. As shown in Supplementary Fig. 13, we confirmed that neither ecarin, textilinin, nor when combined had any effects on cell viability for a panel of cells, including human fibroblasts, endothelial cell line (EA.hy926), human keratinocyte cell line (HaCaT), and mesenchymal stem cells (MSCs), supporting the previous demonstration of the systemic and cellular biocompatibility of these proteins[43].

## Fibrin 3D architecture controls in vitro fibroblast differentiation

We proposed that the biophysical properties of the provisional fibrin matrix play a role in healing outcomes by regulating fibroblast-to-myofibroblast differentiation, which is known to lead to fibrosis-related ECM production[44]. To validate this hypothesis, we chose the

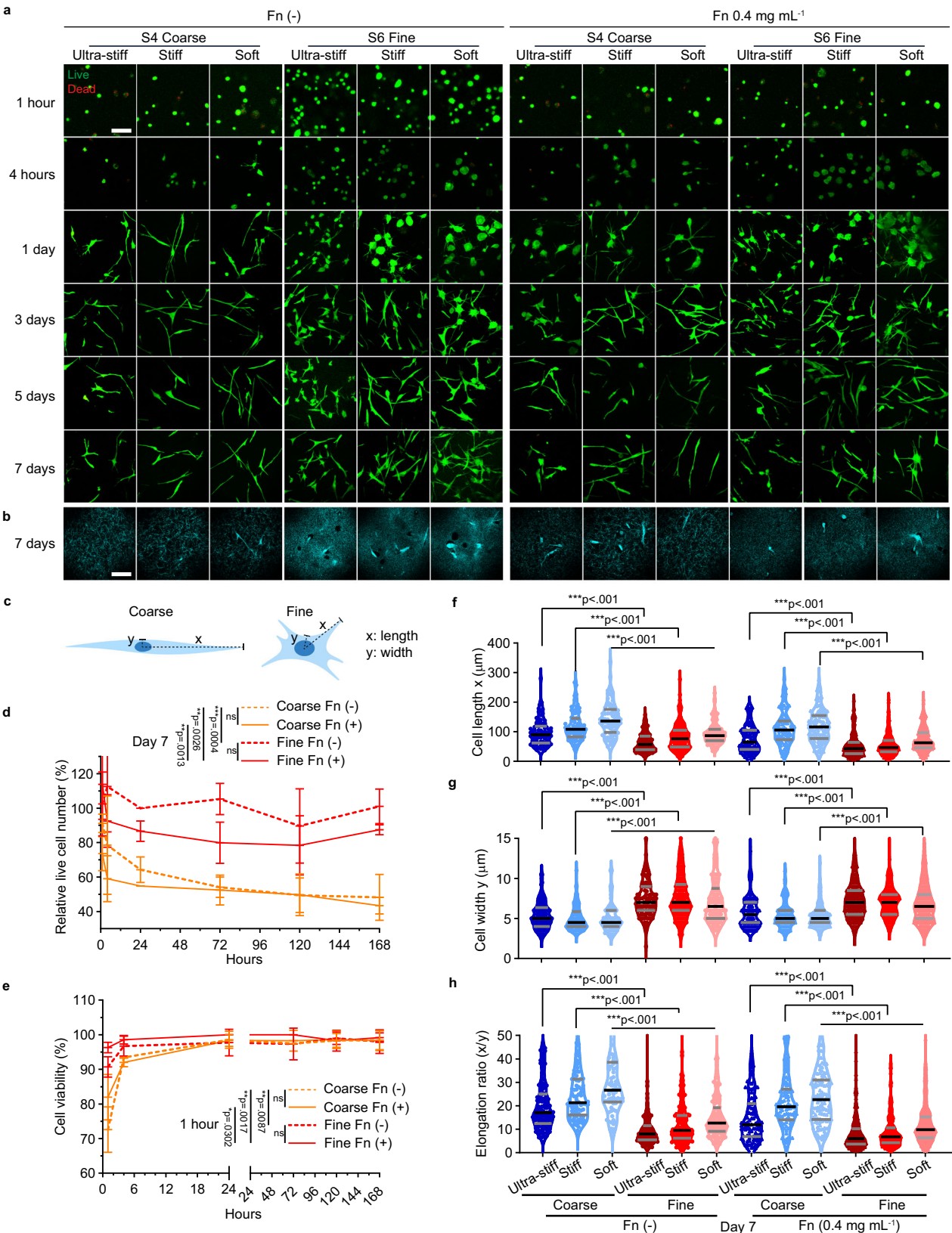

four aforementioned networks (S4 and S6; stiff and soft; without fibronectin addition) and detected how fibrin architecture, along with stiffness, influences fibroblast differentiation. Differentiation of fibroblasts under these four conditions was monitored by either expression of the characteristic smooth muscle actin-alpha (α-SMA) or myofibroblast-specific expression of *TAGLN* and *LRRC17*, which have

been previously verified as potential gene markers in tissue-derived fibroblast/myofibroblast subpopulations[45,46]. The biochemical inducer TGF-β1 was added to trigger the differentiation for 7 days. After 3 days of treatment with TGF-β1, cells in the soft coarse network exhibited a slightly higher α-SMA staining and a 2–3-fold upregulation of both *TAGLN* and *LRRC17* expression, suggesting a pro-differentiated status

**Fig. 4 | Influence of fibrin architecture, stiffness, and ligand density on fibroblast spreading and survival. a** Fibroblast viability in different fibrin networks with/without the addition of purified fibronectin (Fn, 0.4 mg mL⁻¹) was detected by live/dead cell viability assays. Confocal microscopic images of fibroblasts in different fibrin networks (S4 as coarse network, S6 as fine network) stained with calcein AM (green, indicating live cells) and ethidium (Red, indicating dead cells) at different time points are shown. In each image, a maximum image projection of a 380 μm × 380 μm area with 100 μm thickness (2 μm/frame) is shown. Scale bar, 100 μm. **b** Single z-frame of confocal images on day 7 indicating both the unlabeled fibrin networks and embedded fibroblasts were shown under the reflectance mode (380 μm × 380 μm). **c** Schematic of characteristic morphology of fibroblasts in coarse and fine networks. Cell length (x, defined as the maximum length from the center of the nuclei to the farthest cell membrane that each cell spread), cell width

(y, minimum length from the nuclear center to the cell membrane), and elongation ratio (x/y) are depicted. **d** Live cells with calcein staining were counted as the percentage relative to the cell number of fibroblasts in the Fn (–), S6, stiff network on Day 1 and recorded from 1 hour to 7 days after encapsulation. **e** Cell viability (percentage of live cell number out of total cell number) in different conditions was monitored for 7 days. In **d**–**e**, data from 3 biologically independent experiments are shown as mean value ± SD (*n* = 3). Two-way ANOVA analysis with Tukey's correction was used to analyze the differences on day 7 in **d** and 1 hour in **e**. **f**–**h** Morphological analysis of fibroblasts grown in different fibrin networks. Violin plots display the cell length, width, and elongation ratio of individual cells in different conditions (day 7, data merged from 3 biologically independent experiments (*n* = 3), two-tailed Wilcoxon signed-rank test). The black line indicates the median value, while the grey lines indicate quartiles. In **d**–**g**, ns not significant. *$p$ < 0.05, **$p$ < 0.01, ***$p$ < 0.001.

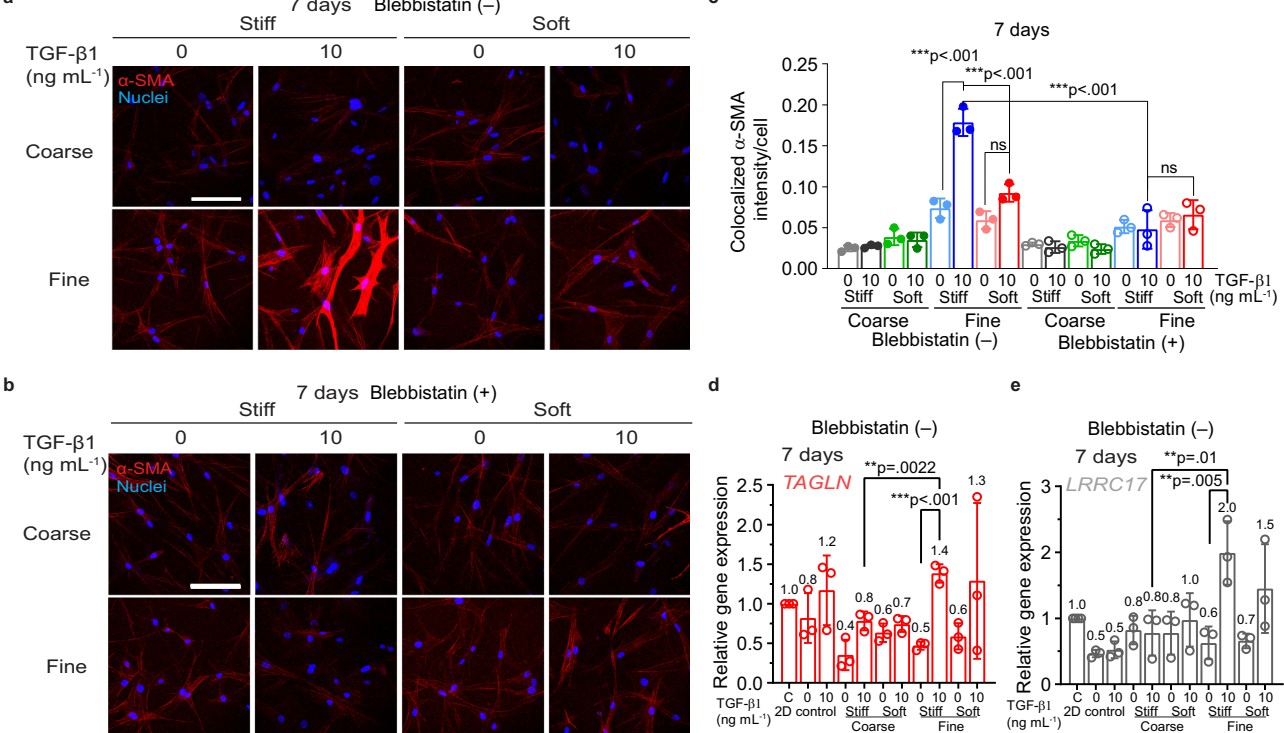

**Fig. 5 | Fibroblast long-term differentiation in coarse and fine fibrin networks.** Representative confocal images of α-SMA immunostaining of fibroblasts encapsulated in different fibrin networks with or without TGF-β1 treatment for 7 days. Cells were treated with 2 μM blebbistatin (**b**) or without blebbistatin (**a**). A maximum image projection (290 μm × 290 μm area with 40 μm thickness) is shown for each image. Scale bar, 100 μm. **c** Quantification of the colocalized α-SMA expression of fibroblasts in different networks. Relative expression of the myofibroblast-specific

genes *TAGLN* (**d**) and *LRRC17* (**e**) in fibroblasts after 7 days of growth in different networks. Dissociated fibroblast pellets on day 0 (denoted as C in the figure) or the fibroblasts cultured on 2D substrates served as control groups. Data in **c**–**e** obtained from 3 biologically independent measurements are shown as mean value ± SD with statistical analysis performed by two-way ANOVA (*n* = 3) followed by Tukey's correction. ns not significant. *$p$ < 0.05, **$p$ < 0.01, ***$p$ < 0.001.

compared to the cells in other conditions (Supplementary Fig. 14). Interestingly, this status was only transient and was reversed after extended culture (by day 5). After 5–7 days, the fibroblasts in the stiff fine network exhibited the highest level of α-SMA expression in response to TGF-β1, while the cells in the soft fine network showed medium α-SMA intensity (Fig. 5a, c for 7 days, Supplementary Fig. 14b, c for 5 days). In comparison, the fibroblasts in the coarse networks showed a much lower level of α-SMA despite the presence of TGF-β1. Notably, cells in the fine networks, in the absence of TGF-β1, exhibited a medium level of α-SMA stress fibers, suggesting the critical role of the fine architecture on fibroblast differentiation. This observation was supported by a 2-fold increase in *TAGLN* and *LRRC17* gene expressions in these cells compared to the cells in the coarse networks in response to 7 days of treatment with TGF-β1. Meanwhile, we detected collagen expression of these differentiated fibroblasts in

different fibrin networks, which signals the functional deposition of fibroblasts. As shown in Supplementary Fig. 15, collagen expression had the highest deposition in the stiff fine fibrin in response to TGF-β1. Interestingly, fibroblasts cultured in 2D did not show characteristic α-SMA fibers after 7 days of TGF-β1 treatment (Supplementary Fig. 16). These observations strongly suggest that a coarse fibrin 3D architecture facilitates a temporal short-term fibroblast differentiation, while a fine fibrin 3D architecture promotes long-term fibroblast differentiation in response to TGF-β1.

### Fibrin 3D architecture regulates in vitro fibroblast mechanotransduction pathway
Biophysical cues have previously been shown to regulate cellular behaviors by focal adhesion assembly and actin cytoskeleton-mediated mechano-sensing and mechano-transduction pathways[47–49].

Therefore, to investigate the underlying mechanisms during the architecture-regulated fibroblast differentiation, we monitored cytoskeleton alignment and the YAP mechanotransduction pathway over 7 days. After 3 days, cells in the coarse network exhibited stronger F-actin fibers, while the cells in the fine network assumed weaker F-actin fibers (Supplementary Fig. 17a, c). Nuclear YAP was negative in all conditions during this early phase (Supplementary Fig. 17d, f, g), suggesting that YAP was not responsible for fibroblasts' early responses to the fibrin matrices. However, this trend was reversed after long-term culture. Fibroblasts encapsulated in the fine networks adopted stronger F-actin fibers (Supplementary Fig. 17b and c for 5 days and Fig. 6a, d for 7 days), and all conditions showed high rates of nuclear translocation, with the highest YAP nuclear intensity observed in the cells embedded in fine networks in the presence of TGF-β1 (Supplementary Fig. 17e, f, g for 5 days and Fig. 6c, e, f for 7 days). Notably, cells in the fine networks containing TGF-β1 also exhibited a high cytoplasmic YAP expression, representing a large potential repository to reinforce nuclear translocation. These results imply that a fine network facilitates long-term fibroblast F-actin alignment and YAP mechanotransduction pathway activation during the fibroblast-to-myofibroblast differentiation.

Since fibroblast-to-myofibroblast differentiation requires cytoskeleton contractility-induced nuclear chromatin remodeling through the linker of nucleoskeleton and cytoskeleton (LINC) complex[50–52], we speculated that cytoskeleton contractility could play a role in the architecture-regulated fibroblast differentiation. We depleted cytoskeleton contractility with the addition of myosin II inhibitor blebbistatin. We found that fibroblasts failed to exhibit enhanced α-SMA (Fig. 5b, c) or F-actin (Fig. 6b, d) fibers in the stiff fine network in response to TGF-β1. This result suggests that the architecture-regulated fibroblast differentiation is dependent on cytoskeleton contractility. Lastly, we detected focal adhesion formation in different fibrin networks, which physically links integrins to intracellular actin cytoskeleton structures and mediates force transmission between ECM and cells[53,54]. We found that after 7 days of TGF-β1 treatment, fibroblasts in the stiff networks exhibited a strong expression of vinculin clusters, a component of focal adhesion complexes as compared to the weaker vinculin expression in soft fibrin networks (Supplementary Fig. 18a–c). These data suggest that focal adhesion formation also plays a role during the architecture and stiffness-regulated fibroblast differentiation.

Taken together, as schematically outlined in Fig. 6g, our data demonstrate that the 3D fibrin architecture alone can strongly influence early fibroblast survival, spreading, and long-term proliferation; it also controls long-term fibroblast differentiation in matrix stiffness and cell contractility-dependent manners.

## Discussion

(Bio)synthetic hydrogels have been widely developed in the last few decades to mimic in vivo ECM due to the disadvantages of natural hydrogels, such as uncontrollable architectural and mechanical properties and batch-to-batch variation. This work overcomes these common limitations by introducing a protocol to form natural fibrin hydrogels with precisely controlled architectural and mechanical properties suitable for long-term 3D cell culture. This snake venom-controlled fibrin system allows independent control of pore size (1–13 μm) and stiffness (0.2–1 kPa), providing a defined platform for biophysical studies. By characterizing and tuning the FXIII activity in the fibrinogen, we also identified and overcame batch-to-batch variation of fibrin matrices. The precise control of the enzyme-catalyzed self-assembled fibrin matrices allows investigation of the role of 3D architecture and matrix stiffness on cellular responses without changing the integrin ligand concentration. Human dermal fibroblast behaviors, including early cell survival, cell spreading, cytoskeleton alignment, and differentiation, were shown to be influenced by a subtle change in the material's pore size (~2 μm). This adds to the current biophysical

theory of the mechanical influence on cellular responses and the sensitivity to biochemical cues. Moreover, we demonstrate that a subtle change in matrix stiffness (~200 Pa) is enough to alter these cellular responses significantly. In the short term (<3 days), fibroblasts spread faster in the soft coarse networks in a YAP-independent manner. We attribute this behavior to the 3D matrix confinement theory[55]. Different from the 2D culture, where cells grow on a substrate, cells cultured in the 3D matrices are dominated by spatial confinement, governed by matrix pore size, degradability, and viscoelastic properties. Fibroblasts in these networks overcome less elastic and larger pores, spreading quicker in the softer coarse fibrin networks; this early spreading-induced response is YAP independent as no YAP nuclear translocation was observed and likely related to the spreading-dependent RhoA-ROCK or ion-channel pathways[39,56,57] as cells spread faster in the softer coarse networks. This response, however, cannot fully explain the early cell loss in the coarse networks. We speculate that this cell loss is due to the loss of cell attachment, termed anoikis[58] in coarse networks, where the integrin ligands are less accessible (~30% decrease as compared to fine networks, as quantified by specific surface area, Supplementary Fig. 2c). This observation would explain why the fibroblasts are less sensitive to TGF-β1 in coarse networks. Fibroblasts in coarse networks have fewer ligands to attach to and hence weaker integrin-dependent mechanotransduction, which is insufficient to enable LINC complex-mediated nuclear mechano-sensing during fibroblast differentiation[59]. For the cells in coarse networks, focal adhesion-mediated F-actin stress fibers are not strong enough to activate cell contractility-induced nuclear mechano-sensing and mechanotransduction[49]. This effect is further enhanced when the stiffness is reduced to 200 Pa as cell contractility is also matrix stiffness dependent[60]. Therefore, cells in the coarse and soft matrices form less focal adhesion complexes (Supplementary Fig. 18) and weaker F-actin stress fibers (Fig. 6) and undergo less contractility-mediated mechano-sensing and mechanotransduction.

The significance of this study lies in generating a tunable biomaterial to enable reproducible investigations of natural 3D cellular functions, with control over pore size and biomechanics not previously reported (procedures concluded in Supplementary Fig. 19g). The synergistic effects of fibrin architecture and mechanics on fibroblast survival, cell spreading, and differentiation also have vital implications for wound healing. Recent innovations in scar modulation have shifted their focus to mechanobiology[16,17,61]. Following this trend, our study indicates a promising biophysical strategy to achieve scarless wound healing – inducing a coarse and soft network to reduce the profibrotic myofibroblasts by either inducing cell death or inhibiting differentiation. We expect that this snake venom-controlled fibrin system may fuel approaches to developing smart biomaterials for tissue engineering and regenerative medicine.

## Methods
### Materials
The research in this study complies with all relevant ethical regulations and is approved by The University of Queensland, Engineering, Architecture and Information Technology, Low and Negligible Risk Ethics Sub-Committee and Australian Red Cross Lifeblood Ethics Committee.

Ammonium sulfate, HEPES, sodium citrate, Tris base, sodium chloride, DEAE-Sepharose resin (CL-6B, DCL6B100), and D₂O were purchased from Sigma-Aldrich. 7000 MWCO dialysis membrane (SnakeSkin Dialysis Tubing, 7K MWCO) was purchased from Thermo-Fisher Scientific. 30k filter tubes (Amicon ULTRA-15 15 mL 30 K centrifugal filter unit) were purchased from Merck. SDS-PAGE 5× loading buffer consists of 50% Glycerol (#56-81-5, Sigma-Aldrich), 10% 2-Mercaptoethanol (#60-24-2, Sigma-Aldrich), 10% Sodium Dodecyl Sulfate (#151-21-3, Sigma-Aldrich), and 5% Trizma Hydrochloride (#1185-53-1, Sigma-Aldrich).

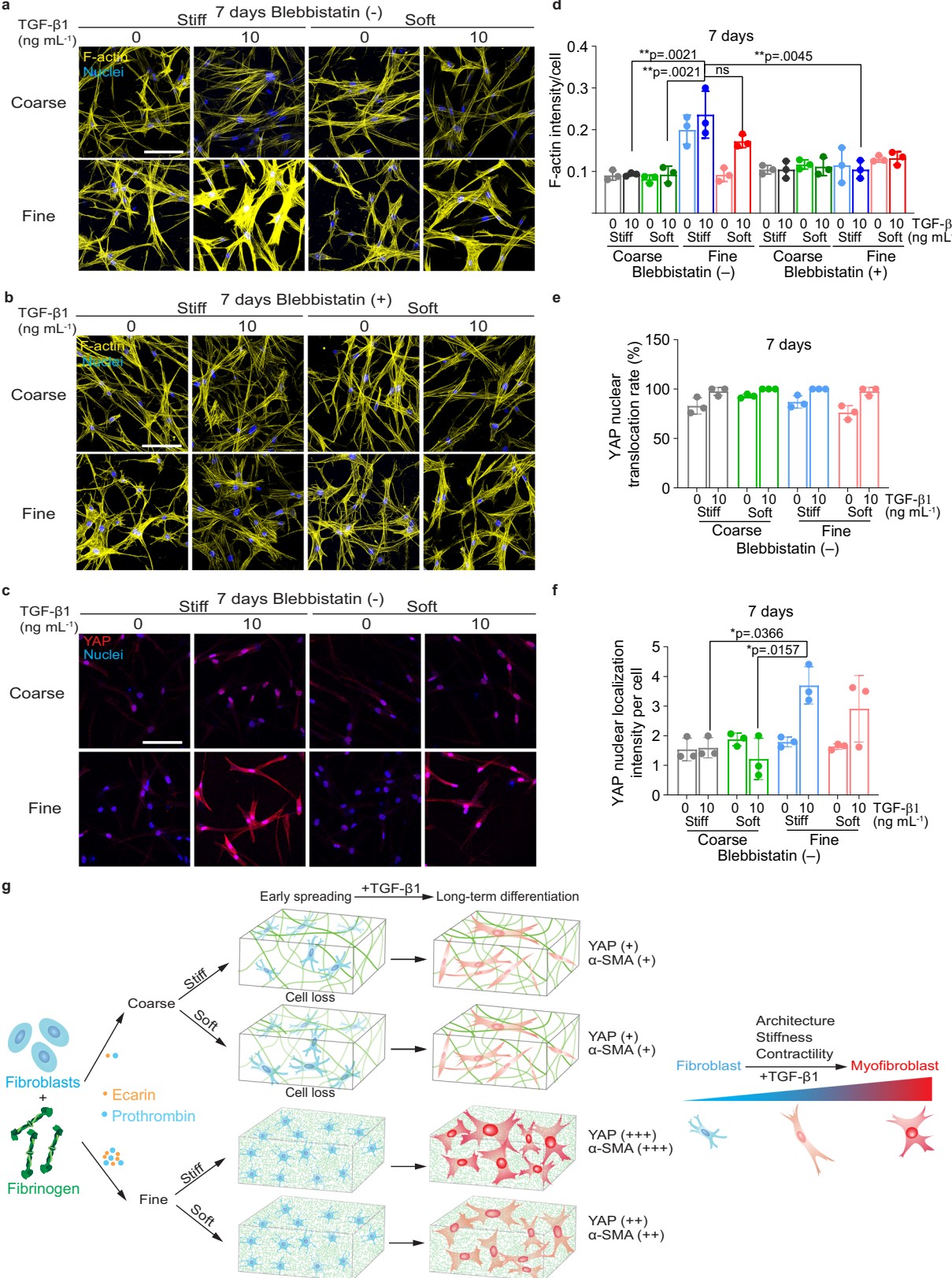

Human blood was obtained from Australian Red Cross Lifeblood, Blood bank, Queensland (human ethics approval #2018001922). Provided as part of a Material Supply Deed: 20-09QLD-05 with The Australian Red Cross Lifeblood, where donors provided informed consent.

Human prothrombin (#HCP-0010), human alpha-thrombin (#HCT-0020), and FXIII standard protein (HCXIII-0160) were purchased from Haematologic Technologies, USA. 1, 3-Dimethyl-4, 5-diphenyl-2-[(2-oxopropyl)thio] imidazolium trifluorosulfonic acid salt (D004) was purchased from Zedira, Germany. TGF-β1 (#240-B-002) was purchased from R&D SYSTEMS, USA. Recombinant textilinin (#Q8008-1105) was generously provided by Kong-Nan Zhao and Martin Lavin. Recombinant ecarin was generously provided by Q-SERA Pty Ltd. Anti-YAP primary

**Fig. 6 | Fibroblast F-actin expression and YAP nuclear translocation in coarse and fine fibrin networks.** Confocal images of F-actin staining after fibroblasts were encapsulated in different fibrin networks for 7 days. Cells were either treated with 2 μM blebbistatin (**b**) or without blebbistatin (**a**). **c** Confocal images of YAP staining 7 days after encapsulation. All the images in **a**–**c** are shown as a maximum image projection (290 μm × 290 μm area with 40 μm thickness). Scale bar, 100 μm. Quantification of F-actin intensity (**d**), YAP nuclear translocation rate (**e**), and YAP nuclear intensity (**f**) over 7 days. Data in **d**–**f** were obtained from 3 biologically independent measurements and are shown as mean value ± SD ($n = 3$). Statistical analysis in **d** and **f** was performed by two-way ANOVA with Tukey's correction. ns, not significant. *$p < 0.05$, **$p < 0.01$. **g** Illustration of fibroblast early spreading and long-term differentiation in different networks in response to TGF-β1. Fibrin fibers are depicted in green. Cell differentiation status is illustrated by colors according to the α-SMA level (blue as undifferentiated status, yellow as pro-differentiated status, and red as differentiated status). Fibroblasts adopt a spread morphology in coarse networks after early spreading (<3 days), while cells in the fine networks spread less. Cells spread more in the soft than in the stiff networks for either coarse or fine networks. During this time, YAP nuclear translocation is negative (shown as blue nuclei). After long-term growth (5–7 days), fibroblasts in the fine networks exhibit a spread morphology. Only the cells in the stiff fine network have a strong α-SMA expression, indicating a fully differentiated status. The nuclear YAP intensity is also strongest in the stiff fine network depicted as red nuclei, while it has a relatively lower intensity in the soft fine network and the coarse networks depicted as yellow nuclei.

antibody (#sc-101199) was purchased from Santa Cruz Biotechnology, USA. Anti-α-SMA primary antibody (#A2547), CellMask Deep Red Plasma Membrane Stain (#C37608), and blebbistatin (#B0560) were purchased from Sigma-Aldrich. Fibronectin Rabbit mAb Alexa Fluor 647 conjugate (#72943) was from Cell Signaling Technology. Anti-Vinculin Mouse mAb (#V9131) was obtained from Sigma-Aldrich. Anti-GAPDH Rabbit polyclonal Ab (#10494-1-AP) was from Proteintech. Goat anti-Rabbit HRP-conjugated Ab (#AQ132P) and Rabbit anti-Mouse HRP-conjugated Ab (#AQ160P) were purchased from Merck. pET-28a-mCherry-CNA35 (Plasmid #61603) was obtained from Addgene.

Chromogenic FXIII activity assay (Technochrom, #5360010) was purchased from Diapharma. Molecular probes Dextran conjugates (10k MW, Alexa Fluor 647, #D22914, and 70k MW, Texas Red, #D1864) were purchased from Invitrogen, ThermoFisher Scientific. 4–20% Tris-Glycine Mini Gels (#XP04205BOX) and Trans-Blot Turbo PVDF Transfer packs (#1704156) were from Bio-Rad. Cell counting kit-8 (CCK-8, #96992) was purchased from Sigma-Aldrich.

Alexa Fluor Plus 647 Phalloidin (#A30107), Hoechst 33342 (#H1399), Goat anti-Mouse Alexa Fluor Plus 594 secondary antibody (#A32742), Alexa Fluor 488-conjugated fibrinogen (Molecular Probes, Inc) and the protein ladders (#26614, #26625), Dulbecco's Modified Eagle Medium (DMEM, #21063029), fetal bovine serum (#10099141), penicillin-streptomycin (#15140122), LIVE/DEAD Viability/Cytotoxicity kit (#L3224) were all purchased from ThermoFisher Scientific.

The 15-well chamber slides (#81506) were purchased from Ibidi, USA. Chromatography columns (Poly-Prep, #731-1550) were bought from Bio-Rad. Cell culture inserts (3 μm filter, #MCSP24H48) and 24-well plates (#CLS3524) were purchased from Sigma-Aldrich. Triton X-100 was purchased from Sigma-Aldrich. RNeasy kit (#74106) was purchased from QIAGEN. TRIzol reagent (#15596026), Reverse Transcriptase kit (#18080093), TaqMan Universal PCR Master Mix (#4364340), and TaqMan gene expression assays including *TAGLN* (Hs01038777_g1), *LRRC17* (Hs00180581_m1), *GAPDH* (#4326317E) and *18S rRNA* (#4319413E) and the polystyrene flasks (Nunc™ EasYFlask™) were all purchased from ThermoFisher Scientific. Chromogenic substrate assay (S-2238) was purchased from Diapharma, USA.

## Fibrinogen preparation

Fresh human citrated whole blood (300–500 mL per donor) was centrifuged at room temperature (RT), 1500 $g$ for 15 min, to obtain platelet-poor plasma (PPP). The PPP was further centrifuged at RT, 5000 g for 15 min to obtain plasma without platelets. The individual plasma (from at least 3 donors) depleted of platelets was pooled and hereafter referred to as plasma.

Ammonium sulfate (AS) precipitation was employed twice to purify the fibrinogen from the plasma. For the first AS precipitation, plasma and ammonium sulfate were chilled on ice. Plasma was precipitated by 0.9 M AS containing 10 mM HEPES, pH 7.4. The resulting solution was swirled in a plastic beaker at 4 °C for 2 hours to precipitate the fibrinogen. The precipitated solution was subsequently centrifuged at 4 °C 5000 $g$ for 15 min and resuspended in resuspension buffer (20 mM sodium citrate, 20 mM Tris pH 6.8) at RT for 30 min to

fully dissolve the pellet for the second AS precipitation. A second AS precipitation was employed, and the resulting pellet was resuspended in fibrinogen buffer (10 mM HEPES pH 7.4, 10 mM sodium citrate, 150 mM sodium chloride) for 30 min until fully dissolved. The precipitated protein was dialyzed using the 7000 MWCO dialysis membrane to dialyze against the buffer containing 10 mM HEPES, 10 mM sodium citrate, and 150 mM NaCl pH 7.4 at 10 °C for 12 h. Afterward, the anion-exchange chromatography using DEAE-Sepharose was carried out to remove the fibronectin from the precipitated fibrinogen product[38]. A total of 50 mL DEAE-Sepharose resin was loaded into a chromatography system equipped with a 20 cm long column (Radius = 2 cm) and a streamline dispenser (IPC high precision multichannel dispenser, ISMATEC). The sample was applied to the resin at 0.5 mL min⁻¹ after the column was equilibrated with equilibration buffer (10 mM HEPES, 10 mM sodium citrate, 100 mM NaCl, pH 7.4). A two-column volume of washing buffer (10 mM HEPES, 10 mM Sodium Citrate, 100 mM NaCl, pH 7.4) was then loaded, and the washing fractions were collected. The fibronectin was then collected by gradient elution using the elution buffer (10 mM HEPES, 10 mM Sodium Citrate, 150 mM–1 M NaCl, pH 7.4). The collected fractions were monitored by Bradford assays as well as SDS-PAGE during the chromatography. The NaCl concentrations of the fibrinogen and fibronectin fractions were adjusted to 150 mM. Both fibrinogen and fibronectin fractions were then pooled and concentrated using 30k filter tubes. The concentrations of the purified fibrinogen (pFg) and purified fibronectin (pFn) were quantified using a NanoDrop™ 2000/2000c Spectrophotometer (ThermoFisher Scientific) using the extinction coefficient (E$^{1\%}$) of 15.1 (280 nm)[2] for fibrinogen[62] and 14 for fibronectin. Notably, the final purified fibrinogen product exists in the form of the fibrinogen-fibronectin complex, according to the SDS-PAGE results. The final purified fibrinogen or fibronectin was stored at −80 °C. Eight batches of purified fibrinogen were obtained as part of the study (B1–B10). Three continuous fractions of fibrinogen (labeled as a, b, c, or d) were collected for 3 batches (B4a–d, B7a–d, and B8a–c).

## FXIII isolation

To obtain the FXIII in the purified fibrinogen product, fibrinogen was denatured and removed by incubating the product at 60 °C for 3 min. Denatured fibrinogen protein was then removed using a pipette tip. The obtained FXIII was quantified by SDS-PAGE and stored in 10% glycerol at −80 °C.

## FXIII activity quantification

The endogenous FXIII activity in each batch of fibrinogen was quantified by the chromogenic FXIII activity assay according to the manufacturer's instructions.

## Fibrin formation

For the venom-controlled fibrin network formation, frozen fibrinogen was thawed and warmed to RT for 30 min, followed by mixing with a mixture of 2 mM CaCl₂, 10 μM textilinin, human prothrombin ($10^{-7}$–$10^{-3}$ IU μL⁻¹, with 1 unit equivalent to the prothrombin activity in 1 mL normal

human plasma) in a buffer containing 20 mM HEPES and 150 mM NaCl, pH 7.4 (hereafter referred as clotting buffer). This mixture and prepared ecarin in clotting buffer were then pre-heated separately at 37 °C for 5 min and then combined to form the fibrin networks in 15-well chamber slides at 37 °C. For the thrombin-initiated fibrin networks, alpha-thrombin ($10^{-4}$–$10^{-3}$ Unit µL$^{-1}$, determined by fibrinogen clotting assay relative to human NIH standard thrombin) was pre-heated for 5 min and added to the fibrinogen in the clotting buffer under the same condition. For the fibrin networks involving tuning the rheological properties, purified FXIII (0–60 µg mL$^{-1}$) or 1, 3-Dimethyl-4, 5-diphenyl-2-[(2-oxo-propyl)thio] imidazolium trifluorosulfonic acid salt (D004), an inhibitor of FXIII, was added to the mixture before initiation. For the experiments detecting the effects of pre-heating time on the fibrin gelation, thrombin or ecarin/prothrombin were pre-heated for either 5 min or 10 min before initiation. The operation time was controlled to be within 5 min after the reagents were pre-heated before initiation. For tuning the final fibronectin concentrations in the fibrin, different concentrations of the purified fibronectin (pFn) were added to the purified fibrinogen (pFg) before initiation whilst maintaining the fibrinogen concentration unchanged. For quantifying the incorporated fibronectin in the 4 mg mL$^{-1}$ fibrin networks, the purified fibronectin product (pFn-B7 or pFn-B8) was employed using the same method under the same condition as fibrin alone (with and without 0.4 mg mL$^{-1}$ fibronectin). The formed fibronectin networks were referred to as S4-Fn and S6-Fn.

To minimize activity degradation, all the enzymes dissolved in 10% glycerol, including prothrombin, thrombin, and FXIII, were always kept at −80 °C before mixing. Fibrinogen and fibronectin were aliquoted and only freeze-thawed once to avoid the degradation of the activity of endogenous FXIII.

## SDS-PAGE
The purity of different fractions during chromatography or concentrated samples was detected by reducing SDS-PAGE. For this, 10 µL of each sample was harvested. Each sample was treated with SDS-PAGE loading buffer and denatured at 95 °C for 10 min. Protein ladders and samples (5 µL per lane) were then loaded on 4–20% polyacrylamide gel. Electrophoresis was run in 1× SDS Running Buffer at 40 mA for 30 min. The gels were stained with Coomassie Blue and scanned by the Che-miDoc Imaging system (Bio-Rad, V6). Densitometry analysis of the bands was performed by ImageJ. For quantification of incorporated fibronectin percentage, purified fibronectin (Fn) before gelation, gel pellet 1 h after gelation, and the remaining supernatant buffer, as well as the gel, were detected by reducing SDS-PAGE. The incorporated fibronectin percentage was calculated as Eq. 1.

$$\left(1 - \frac{Free\,fibronectin\,in\,supernatant}{Total\,amount\,of\,fibronectin}\right) \times 100\% \qquad (1)$$

## Rheology
The rheological properties of the fibrin networks were evaluated and quantified using an Anton Paar MCR-502WESP rheometer (on Rheo-Compass V1.23) equipped with a parallel stainless-steel geometry PP25. Upon fibrin network initiation, 300 µL of the fibrin sample was immediately loaded onto the pre-warmed (37 °C) bottom plate with the gap set to 0.5 mm, followed by trimming of the excess sample around the geometry. To completely avoid evaporation, silicon oil was applied to seal the sample for long-term measurement. The viscoelastic properties of the fibrin networks during polymerization were monitored dynamically by exerting a constant oscillatory shear strain with a maximum amplitude (γ) of 0.5 % and frequency (f) of 0.5 Hz. The storage modulus (G′), loss modulus (G″), and loss factor (tanδ) were monitored continuously for 6–24 hours until stable properties were achieved. The endpoint storage modulus (G′) or loss factor (tanδ) of each network was recorded as the matrix stiffness or loss factor of the individual

network. The gelation lag time of the fibrin network was measured as the time point when G′ of the respective network reached 5 Pa.

## Small and ultra-small angle neutron scattering
To determine the structural properties over the broad range of dimensions of the fibrin network down to the internal structure of the individual fibers, a combination of small-angle neutron scattering (SANS) and ultra-small angle neutron scattering (USANS) techniques were employed.

## Fibrin sample preparation for SANS /USANS measurements and corresponding CLSM
For the D$_2$O-based measurements, purified fibrinogen (B4) was dialyzed against the deuterated buffer containing 20 mM HEPES, 2 mM sodium citrate, and 500 mM NaCl in 90% D$_2$O at room temperature for 12 h and was then used for the SANS/USANS and corresponding CLSM. For the H$_2$O-based SANS/USANS measurements and corresponding CLSM comparisons, the purified fibrinogen (B4) was buffer exchanged by dialysis against the same base buffer, 20 mM HEPES, 2 mM sodium citrate, and 500 mM NaCl but in H$_2$O. The pre-heating time of the reagents before initiation varied from 5 to 10 min. After loading into the SANS/USANS cells/sample holders, samples were incubated at 37 °C for at least 2 hours before the commencement of the neutron scattering measurements to ensure complete network formation.

## SANS/USANS measurements
Both small-angle neutron scattering (SANS, QUOKKA)[63] and ultra-small angle neutron scattering (USANS, KOOKABURRA)[64] measurements were carried out at the OPAL reactor at the Australian Nuclear Science and Technology Organisation (ANSTO, Lucas Heights, Sydney, Australia). SANS measurements on QUOKKA covered a q range from $6\times10^{-4}$ to 0.5 Å$^{-1}$, using the incident beam with wavelengths of 6 Å and 9.63 Å (focusing optics) with a resolution of 10%. q is the magnitude of the scattering vector and is defined as q = 4π sin(θ)/λ, where θ and λ denote half the scattering angle and wavelength respectively. Measurements were conducted at three instrument configurations with source-to-sample (SSD) and sample-to-detector distances (SDD) of SSD = SDD = 20 m (standard and focusing); SSD = SDD = 8 m and SSD = 4 m and SDD = 1.3 m with the latter employing a 300 mm lateral detector offset. The source and sample aperture diameters were 50 mm and 12.5 mm, respectively. Samples were prepared in quartz cuvettes of 2 mm thickness. USANS measurements were conducted on the KOOKABURRA double-crystal diffractometer using a Gd aperture (29 mm in diameter) with a neutron wavelength of 4.74 Å, which extended the q range down to $4 \times 10^{-5}$ Å$^{-1}$. All the measurements were conducted at 37 °C. SANS data were reduced to absolute values, and the reduced slit-smeared USANS data were 'desmeared' using the Lake algorithm before merging with the QUOKKA data[65,66]. Combined SANS and USANS scattering curves were plotted after subtracting the incoherent background. All fitting and analysis were performed using the SASView software Version 4, instructed by other fundamental works on the neutron scattering analysis of fibrin gels[31,67]. Scattering length density (SLD) was calculated using a protein neutron scattering length density calculator program (Science and Technology Facilities Council (STFC)[68]. Specifically, the average fiber radius (R) was calculated using Guinier analysis according to Eqs. 2 and 3, where I(Q) is the scattering intensity at scattering vector (Q) and Rg is the radius of gyration.

$$I(Q) = \frac{I(0)}{Q}\exp\left(-\frac{Q^2 R_g^2}{2}\right) \qquad (2)$$

$$R = Rg \times \sqrt{2} \qquad (3)$$

The volume fraction of the internal protofibrils (φ$_{Int}$) was extracted by invariant analysis using Eq. 4, where Q* is the invariant of the

high q Porod region. $\varphi_{Fib}$ is the volume fraction of the total fibrinogen protein and was calculated as 0.003[69].

$$Q^* = \int_0^\infty q^2 I^*(q) dq = 2\pi^2 \varphi_{Fib}(1 - \varphi_{Int})(\triangle\rho)^2 \qquad (4)$$

The specific surface area ($S_v$) was calculated using Eq. 5, where $C_P$ is the Porod constant.

$$S_v = \frac{\pi\varphi_{Fib}(1 - \varphi_{Fib})C_p}{Q^*} \qquad (5)$$

The average number of protofibrils per fiber (Np) was calculated using Eq. 6, where $\mu_{Fiber}$ is the mass-length ratio of a fiber; $\mu_P$ is the mass-length ratio of a protofibril (340 kDa / 22.5 nm); $\rho$ is the density of fibrinogen protein (1.395 g mL$^{-1}$), and R is the average fiber.

$$Np = \frac{\mu_{Fiber}}{\mu_P} = \frac{\varphi_{Int}\rho R^2}{\mu_P} \qquad (6)$$

The fractal dimension ($D_f$) and correlation length ($\xi$) were extracted according to Eq. 7, where S(q) is the structure factor.

$$S(q) = 1 + \frac{3(\sin(qR_0) - qR_0\cos(qR_0))}{\left[1 + \frac{1}{(q\xi)^2}\right]^{\frac{D_f}{2}}} \qquad (7)$$

## Cell growth and 3D cell culture

Cell experiments were performed under the University of Queensland human ethics (Approval number: 2019000588). Human fibroblasts (#PCS-201-012, ATCC), EA.hy926 (#CRL-2922, ATCC), keratinocytes HaCat (obtained from Translational Research Institute, Australia), and mesenchymal stem cells (obtained from StemCore, Australia) were cultured in Dulbecco's Modified Eagle Medium containing 10% fetal bovine serum (FBS) and 1% (v/v) penicillin-streptomycin on the polystyrene flasks.

For 3D cell culture, $5 \times 10^5$ mL$^{-1}$ (for cell viability detection) or $10^6$ mL$^{-1}$ (for cell differentiation experiments) fibroblasts dissolved in PBS were added to the initiated fibrin mixture 1–5 min before the gelation of the respective network according to the gelation lag time. The mixture was then loaded to pre-warmed 15-well chamber slides (10 μL/well) and quickly transferred to a 37 °C incubator, flipping the chamber slides gently every 30 s until network formation was observed to avoid cell sinkage. The same number of cells were prepared in different wells. Thirty minutes after the network was formed, complete media (containing 10 μM textilinin) was added to each well. The media was changed after 4 h of incubation and subsequently every 2 days. For the induction of fibroblast differentiation, 5–10 ng mL$^{-1}$ recombinant human TGF-β1 was added to the media for up to 7 days. For the RNA extraction experiments, cells were cultured in 96-well plates (100 μL/well) for 5 wells per condition and maintained as described above. For the treatment of blebbistatin, 2 μM was added to the medium during the 7 days culture. For the addition of pFXIII or D004, 0–60 μg mL$^{-1}$ purified FXIII or 0–40 μM D004 were added to the fibrin mixture only before initiation.

## Immunofluorescence staining

Cells were fixed in 4% paraformaldehyde (PFA) for 1 h and permeabilized using 0.1% Triton X-100 for 45 min. For α-SMA and YAP staining, cells were blocked with 1% BSA in PBST buffer for 2 h, before they were incubated with anti-α-SMA primary antibody (1:200) or anti-YAP primary antibody (1:200) overnight. Cells were then incubated with goat anti-mouse Alexa Fluor Plus 594 secondary antibody (1:1000) for 4 h. Cells were also incubated with Alexa Fluor Plus 647

Phalloidin (1:1000) and Hoechst 33342 (1:1000) for 2 h before confocal microscopy imaging. For cell membrane staining, samples were treated with CellMask Green Plasma Membrane Stain (1:1000) for 30 min. The quantification of the immunostaining was performed on the BioImageXD 1.0 platform. For α-SMA expression, only the α-SMA intensity colocalized with F-actin was analyzed to specifically quantify the α-SMA stress fibers. For the YAP quantification, both the nuclear localization rate and nuclear intensity were analyzed. The results were presented as total colocalized α-SMA or nuclear YAP intensity per cell. For the immunostaining of fibronectin incorporated in the fibrin networks formed in 15-well chamber slides, fluorescently labeled (using 1:100 Alexa Fluor 488-conjugated fibrinogen in the purified fibrinogen) S4 or S6 fibrin networks (with 0 or 0.4 mg mL$^{-1}$ fibronectin) were fixed in 4% PFA for 1 h and washed with TBST for 30 min. Fibrin networks were blocked with 1% BSA in PBST buffer for 2 h and incubated with anti-fibronectin fluorescent conjugate (1:100) overnight for further imaging. The colocalization analysis of fibrinogen and fibronectin expressions was done using the BioimageXD V1.0 platform[70]. For the immunostaining of focal adhesions expression in fibroblasts in different fibrin networks, cells after 7 days of treatment of TGF-β1 were co-stained with vinculin antibody (1:200) and Phalloidin F-actin conjugates (1:1000) using the same immunostaining methods described above. For the staining of the total collagen production, collagen binding protein CNA35-mCherry was prepared and purified using the pET-28a-mCherry-CNA35 transformed E.coli BL21 system as described previously[71]. After samples were fixed with 4% PFA washed with PBST, CNA35-mCherry (1:1000) was added and treated for 4 h at RT. A thorough wash with PBST was performed to remove the background. A fibrin network S4 or S6 without cells served as a negative control.

## Confocal laser scanning microscopy (CLSM)

For the architectural characterization of the fibrin networks, Alexa Fluor 488-conjugated fibrinogen was mixed with unlabeled purified fibrinogen with a volume dilution ratio of 1:100 (equivalent final concentration of 0.015 mg mL$^{-1}$). 2 h after fibrin network formation, 3D images were obtained using an inverted Confocal Laser Scanning Microscope (TCS SP8 WLL, Leica Microsystems) equipped with a 40× (HC PL IRAPO 40×/1.10 W CORR) water immersion objective. For each image, a 290 μm × 290 μm × 40 μm (x, y, z) area (1 μm/frame) was imaged, starting at a 20 μm distance from the coverslip surface. For the imaging of immunofluorescent staining, a 290 μm × 290 μm × 100 μm area (2 μm/frame) was imaged within 6 hours after the completion of immunostaining. Each sample had at least 3 images chosen randomly in different areas of the same sample. For the cell images, the fibrin networks were visualized using the reflectance mode (Excitation: 576 nm). For the imaging of the fibroblasts on the 2D plate, a 290 μm × 290 μm × 20 μm area (0.5 μm/frame) was imaged starting from the bottom of the plate. For the Live/Dead cell viability imaging, either 380 μm × 380 μm × 100 μm (2 μm/frame), or 775 μm × 775 μm × 200 μm (2 μm/frame) areas were imaged with a 40× water immersion or 20× (HC PL FLUOTAR L 20×/0.40 DRY) air objective. For the quantitative analysis of the pore size of the fibrin networks, pore, and fiber diameter were measured manually on the Leica platform LAS X (V3.6.0). At least 40 pores and 20 fibers in different regions were measured to determine the average pore size and radius and associated standard deviations. For the imaging of fibronectin incorporated in the fibrin network, three 40 μm × 40 μm randomly picked regions were imaged for each condition. For the imaging of the focal adhesions, immunostained vinculin and F-actin were imaged with a 100× oil immersion objective. At least three randomly chosen 50 μm × 50 μm areas were imaged for each condition. For the imaging of collagen production using CNA35-mCherry, randomly picked 290 μm × 290 μm areas with 40 μm thickness were imaged.

## RNA extraction and qPCR

Fibrin pellets along with encapsulated cells, were collected in 1.5 mL Eppendorf tubes for RNA extraction, after which the pellets were homogenized using pellet pestles. Total RNA was extracted and purified by TRIzol reagent and RNeasy kit according to the manufacturer's protocol. The extracted RNA was then quantified by a NanoDrop spectrophotometer (NanoDrop 2000, ThermoFisher Scientific) and reverse transcribed to cDNA using the Reverse Transcriptase kit. For real-time qPCR, TaqMan Universal PCR Master Mix predesigned TaqMan gene expression assays were used to detect *TAGLN* and *LRRC17*. *GAPDH* or *18S rRNA* was used as an endogenous control. qPCR was performed in the CFX Real-Time PCR Detection System (Bio-Rad). Relative gene expression levels were analyzed by the $2^{(-\Delta\Delta Ct)}$ method. Results from the 2D cell pellet before encapsulation served as the control for other samples.

## Spectrophotometry

To gain information about the kinetics of the fibrin network formation, fibrin networks were initiated in 96-well plates with (100 μL/well), and the absorbance at 582 nm was continuously monitored for 1 h at 37 °C in a microplate reader (Tecan M200 Pro Infinite, using i-control software V1.1). To monitor the long-term stability of the fibrin networks, the samples were covered with silicon oil to avoid evaporation and incubated at 37 °C. The absorbance values of the fibrin samples were then recorded every 24 hours. Multiple regions were recorded for each well, and the mean values were determined as the final absorbance readings.

## Western blot analysis

To quantify the protein expression levels of focal adhesion-related protein vinculin and GAPDH in fibroblasts, western blot analysis was performed. When preparing fibroblast samples in different fibrin conditions, fibroblast samples embedded in S4-stiff, S4-soft, S6-stiff, and S6-soft fibrin networks after 7 days of treatment with TGF-β1 were obtained after detaching the fibrin matrix from 96-wells (100 μL per well, 5 wells per condition). Fibrin pellets, along with cells, were collected in 1.5 mL Eppendorf tubes and lysed in RIPA buffer. After denaturing protein samples with 1× loading buffer at 95 °C for 10 min, 10−20 μg (in 5 μL) of protein of each sample and 5 μL protein markers were loaded on 4−20% Tris-Glycine Mini Gels for SDS-PAGE. Protein on the gels was then transferred to PVDF membranes using the Trans-Blot Turbo Transfer System (#1704150, Bio-Rad). After blocking with TBS + 1% BSA buffer, membranes were treated with anti-vinculin and anti-GAPDH primary antibodies (1:1000) for 2 hours at RT. After washing using TBST buffer, membranes were treated with HRP-conjugated secondary antibodies (1:5000) for 1 hour at RT. After washing using TBST buffer, imaging of blots was obtained with the ChemiDoc MP imaging system (Bio-Rad, V6). For semi-quantitative analysis of blots, densitometric readings of band intensities were obtained using FIJI ImageJ software (1.52p). The protein expression level was quantified as the intensity relative to GAPDH and expressed as fold change relative to the S4-Soft group.

## Chromogenic activity assay

A chromogenic substrate assay was used to monitor the fibrinogen cleavage activity by thrombin or prothrombin/ecarin. The substrate was mixed with the clotting buffer with a final concentration of 1.6 mM. Thrombin or prothrombin/ecarin were pre-heated at 37 °C for 5 min or 10 min before initiation. After the substrate was initiated in the 96-well plates with a volume of 100 μL/well, the cleavage activity was monitored for 3 h at 37 °C by detecting the absorbance reading at 405 n subtracted by reference reading at 490 nm ($A_{405-490nm}$) on a microplate reader (Tecan M200 Pro Infinite). The cleavage quantity of the enzymes was determined by the absorbance reading $A_{405-490nm}$. The cleavage rate of the enzymes was analyzed by the $\Delta A_{405-490nm}$/min.

A threshold of $A_{405-490nm} = 0.035$ was set, and the time point when absorbance $A_{405-490nm}$ reached the threshold after initiation was detected.

## Live/Dead cell viability assay

Cell viability was detected using the Live/Dead (Calcein-AM/Ethidium) staining according to the manufacturer's instructions. Briefly, cells along with the fibrin networks in the 15-well chamber slides were washed with PBS at 37 °C for 15 min. They were then incubated with 2 μM Calcein-AM and 5 μM Ethidium at 37 °C for 30 min. Cells were washed and stained with Hoechst for 15 min before performing confocal microscopy imaging. Cell viability was calculated as Eq. 8.

$$\frac{Calcein - positive\ cell\ number}{Calcein - positive\ cell\ number + Ethidium - positive\ cell\ number} \times 100\% \tag{8}$$

## Cell morphology quantification

Cell morphology was quantified using the confocal images obtained during live/dead cell viability detection. Both cell length and cell width were measured manually on projected images of the 3D confocal microscopy images (775 μm × 775 μm × 200 μm) on the Leica platform LAS X. Cell length (x) was defined as the linear length between the center of nuclei and the cell membrane that spread the farthest, while cell width (y) was defined as the shortest length between the center of nuclei and the cell membrane. The elongation ratio of each cell was calculated as the ratio of cell length to cell width (x/y).

## Cell viability/cytotoxicity assays

Cell viability/cytotoxicity was detected by Cell Counting Kit-8 (CCK-8). After cells were seeded in a 96-well plate (500 per well in 100 μL complete media) and incubated for 7 days, 10 μL CCK-8 reagent was added to each well. After 2 hours of incubation, absorbance at 450 nm of the media was read using a microplate reader (Tecan M200 Pro Infinite).

## Fibrin permeability and diffusivity assays

Two setups were employed to test the permeability and diffusion capacity of the established fibrin networks, as previously reported[41]. For detecting the permeability coefficient, 150 μL of S4 or S6 fibrin networks were prepared in an adapted chromatography column as described in Supplementary Fig. 9a. 2 h after gelation, DMEM media was added to the column for equilibration. Before testing, 10 mL DMEM media was prepared with 20 μg mL$^{-1}$ 10 kDa and 100 μg mL$^{-1}$ 70 kDa fluorescently labeled dextran conjugates and loaded into the column. The flow rate was then recorded at 37 °C by testing the volume of eluate every 10 min. Eluate at different time points was collected for fluorescence intensity detection. DMEM media was frequently refilled to maintain the hydrostatic pressure. The permeability coefficient $K$ of different fibrin networks was determined from Eq. 9.[41]

$$K = Q/t \times \frac{L \times \eta}{A \times P} \tag{9}$$

where Q/t is the flow rate of the eluate, L is the height of the column from the surface of the media to the bottom of the fibrin gel (8.4 cm), η is the viscosity of DMEM media (0.00096 Pa.s), A is the effective elution area (0.07 cm$^2$), P is the hydrostatic pressure (785 Pa). For testing the diffusion capacity of the fibrin, a setup (depicted in Supplementary Fig. 9d) mimicking the geometry of 3D cell culture in slide chambers was prepared[41]. 50 μL of S4 or S6 fibrin networks were prepared evenly in cell culture inserts with a 3 μm filter membrane. 2 hours after gelation,

250 μL of DMEM media (containing 10 μM textilinin) with 20 μg mL$^{-1}$ 10 kDa dextran and 100 μg mL$^{-1}$ 70 kDa dextran was added to the top of the fibrin gel. Around 1.25 mL of the same media without dextran was added to the 24-well plates outside the cell inserts. The final media levels were kept at the same height. The diffused media in the 24-well plates were collected at different time points.

When testing the permeabilized or diffused dextran (10 kDa and 70 kDa) intensities, 10 μL of collected eluates or diffused media samples at different time points were diluted to 100 μL. The fluorescence intensities were detected by a microplate reader (Tecan M200 Pro Infinite) using the fluorescence mode. Dextran samples with known concentrations were tested simultaneously and served as standard curves.

### Statistical analysis

For the statistical analysis of immunostaining, the mean values relative to the same condition (soft fine network without treatment of TGF-β1) from at least three independent experiments were included and analyzed. For the analysis of gene expression, relative protein expression levels were analyzed with 2D cell pellets served as control groups. For the analysis of protein expression levels quantified by semi-quantitative analysis, relative protein expression changes were analyzed. These data were included from at least three independent experiments and analyzed by one-way or two-way ANOVA followed by individual comparisons using Tukey's method by GraphPad Prism v9.0. For the quantification of the immunostaining in 2D control, a two-sided paired t-test was performed. For statistical analysis of the cell number, cell length, cell width, and elongation ratio between different groups, the two-tailed non-parametric Wilcoxon Signed test was performed between different groups. Data in all violin plots were shown with median values (solid lines) and quartiles (dashed lines). Other data were expressed as mean ± standard deviation (SD). Standard deviation was calculated using Microsoft Excel for Microsoft 365 (V16.0.13801.20240). $p < 0.05$ was considered statistically significant with specific p values shown in the figures. ns, not significant.

### Statistics and reproducibility

Except for the results with statistical analysis indicating the reproducibility shown in specific figure legends, representative results obtained from one experiment, including the confocal images in Fig. 1b–c, Fig. 4a–b, Fig. 5a–b, Fig. 6a–c, Supplementary Fig. 7c, Supplementary Fig. 10a, Supplementary Fig. 14a–b, Supplementary Fig. 15a, Supplementary Fig. 16a, Supplementary Fig. 17a–b, d–e, Supplementary Fig. 18a, and SDS-PAGE images in Supplementary Fig. 5a, Supplementary Fig. 7a–b, Supplementary Fig. 19c–f, kinetic results in Fig. 2a, Supplementary Fig. 3, Supplementary Fig. 8a, were selected from at least 3 independently performed experiments with similar results. SANS/USANS data for each condition in Fig. 1d–e and Supplementary Fig. 2 were obtained from 1 biological replicate.

### Reporting summary

Further information on research design is available in the Nature Portfolio Reporting Summary linked to this article.

## Data availability

The authors declare that all the data used in this study are provided within the paper and in the Supplementary Information/Source data file. Source data are provided with this paper.

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

## Acknowledgements

This study was supported by the Australian Research Council Laureate Fellowship (A.E.R.: FL160100139), UQ Research Training Scholarship (Z.W.), Australian Institute of Nuclear Science and Engineering travel support (Proposal Reference No. ZIAIO85341), and Australian Nuclear Science and Technology Organisation (ANSTO) proposal grant No. 8534. We acknowledge Romanthi Madawala, Michael A. Taylor, Naatasha Isahak, Samantha Stehbens, and Kristofer Thurecht for their suggestions and advice on the project. We acknowledge and thank Kong-Nan Zhao and Martin Lavin for generously providing recombinant textilinin and Q-SERA Pty Ltd for providing the recombinant ecarin used in this study. We acknowledge Anthony Duff for useful discussions on material preparation for neutron scattering experiments. We acknowledge Australian Red Cross Lifeblood, Queensland, for providing whole blood for this study (Human ethics approval #2018001922). This work was performed in part at the Queensland node of the Australian National Fabrication Facility. A company established under the National Collaborative Research Infrastructure Strategy to provide nano and microfabrication facilities for Australia's researchers.

## Author contributions

Z.W. and J.L. contributed equally to this work. A.E.R., A.W.K., and J.L. conceived and supervised the project. Z.W. and A.W.K. prepared and characterized the fibrin materials. Z.W., J.L., A.W.K., and P.T. characterized the structural and rheological properties of the fibrin networks. J.L., Z.W., E.P.G., and J.M. conducted neutron scattering experiments and analyzed associated data. Z.W., R.Y., and A.W.K. performed cell experiments and analyzed the data. Z.W. and D.X.Z. performed gene expression experiments. Z.W., A.W.K., J.L., and A.E.R. wrote the manuscript. All authors provided critical feedback and helped revise the manuscript.

## Competing interests

The author declares no competing interests.
