## [Peer Review File · Nature Communications]

Snake venom-defined fibrin architecture dictates fibroblast survival and differentiationREVIEWER COMMENTS

Reviewer #1 (Remarks to the Author):

This paper reports a method for precise control of mechanical properties and porosity of fibrin-based scaffolds by using the snake venom-derived protein, ecarin, and prothrombin. The mechanical and structural properties were characterized by rheometer and the combined analysis of SANS and USANS. Moreover, by considering the endogenous activity as well as concentration of FXIII in fibrinogen purified from human plasma, the authors established the procedure to standardize batch-to-batch variations of the naturally derived enzyme. By selecting appropriate reaction conditions, the mechanical properties and porosity can be independently manipulated, from which they found the dominant role of fibrin architectures in the dictation of fibroblasts survival and differentiation. This work addressed an important issue about biophysical regulation of fibroblasts fates in 3D natural hydrogels and it was clearly demonstrated by their excellent molecular designs and characterization. This reviewer thinks the manuscript is suitable for publication in Nature Communications after addressing to the following minor concerns.

1. Even though the authors supplemented fibronectin to control biochemical cues, the activity of fibronectin is not only dependent on the concentration but also on the conformation as well as the degree of fibril formation. The authors need to discuss this point ideally by using several site-specific monoclonal fibronectin antibodies to exclude possible alteration of biochemical cues in their matrix.

2. The change in the porosity affects the diffusion of nutrients and waste products. Cellular traction forces exerted to the hydrogel matrices might further accelerate their diffusion. If this is the case, it is reasonable to see the effect of blebbistatin. To exclude this possibility, the authors need to evaluate the diffusion of some fluorescent probes, such as labeled dextran, through their scaffolds in the presence and absence of cells.

Reviewer #2 (Remarks to the Author):

The manuscript entitled "Snake venom-defined fibrin architecture dictates fibroblast survival and differentiation" by Z. Wang et al. presents a study on natural fibrin hydrogels, which can be tuned in their architectural and mechanical properties so to produce an efficient and rapid wound sealant. These results can be significant as medical applications are likely and will provide a new way to reduce scar formation. However, I am surprised that another article of the group was published in June 2022 in Advanced Health Care Materials with clear relations with this manuscript, but not cited in this one (DOI: 10.1002/adhm.202200574). As the present study is anyway rather specialised, eventually such a journal might be better suited.

The authors have undertaken a series of complementary measurements to evidence the best network which produced the most promising results. The results are sound and statistical analysis was done with care. So I do not think that additional investigations will modify the conclusions. The methods used for the study are standard measurements, data analysis codes, they are correctly cited and well described. Due to space limitation, the Mat. And Meth. Section is a bit short, but it should be sufficient to allow to reproduce the data.

My main criticism concerns the abstract and introduction and I highly recommend that they should be reformulated. The abstract gives already technical details instead of resuming the aim of the study and the main results and possible applications. The introduction contains a lot of repetitions (the word "differentiation" appears 6 times in a short paragraph), although the scientific background permits a broad overview, which should be formulated in a more precise and attracting way.

Minor points:

- 1) Figure 1b is mentioned after Figure 1f, only. The order should be adapted. I find the sub-figures of figure 1 too small.
- 2) In line 103, 9 conditions corresponding to samples S1 – S9 are evoked, but in line 167 a sample S10 appears, which is not introduced.
- 3) A figure 4 appears two times, but figure 5 is missing.

Reviewer #3 (Remarks to the Author):

The authors have conducted an extensive study highlighting the importance of fibrin stiffness and porosity for its performance as a regenerative scaffold. The importance of fibrin architecture and stiffness in regulating cell growth and differentiation. It has been well-established that a quick gelation makes softer hydrogel, which is tightly packed and not producing much porosity which is essential for cell permeation and growth.

Essentially, the study's novelty is that the ecarin-induced gelation provides stiffness, mechanical properties, and porosity of the 3D architecture of the fibrin scaffold.

In every study/evaluation the authors have compared the fibrin properties upon gelation with the physiological enzyme thrombin and the snake venom-derived Ecarin. However, the authors have used fibroblast as the model cell to evaluate and demonstrate that Ecarin controls the gelation time.

Overall, the study is promising and shows that venom-derived Ecarin is suitable for producing 3-D fibrin architecture for tissue regeneration. However, I have the following specific comments:

1. Generally, growth and differentiation of fibroblast is much easier and not much demanding as compared to other more specialized cells like keratinocytes, neural cells, or cardiomyocytes. Therefore, the use of fibroblasts in this study does not provide much value addition. The use of any one specialized cell as a model to demonstrate the cytocompatibility of Ecarin-induced fibrin could add more value.

2. Another problem is the authors have missed the measurement of the fiber diameter which is important for the spreading of cells. The authors should analyze the focal adhesion using specific markers to demonstrate that the cell spreading is optimal.

3. The confocal microscopy at higher magnification after staining for cytoskeletal actin could give information about the cell spreading to demonstrate if the stiffer fibrin formed using Ecarin supports focal adhesion.

4. In the confocal microscope, the adhered fibroblast are spatially far apart and there is no evidence if the cells can grow and cover the porous structures to form functional tissue with deposition of ECM proteins such as collagen and elastin.

5. The authors have not shown if the cells adhere to the superficial fibrin only, or if it has migrated to the internal strands of the 3-D fibrin architecture. The authors could analyze the 3-D construct using a z-sectioning feature of confocal microscopy.

The authors have systematically collected a lot of data on the fibrin gelation time, fiber stiffness, and that the mechanical properties are improved by replacing thrombin with Ecarin. The comparisons with thrombin and prothrombin etc. (with and without FXIII) with the 3-D fibrin formed using Ecarin, clearly present that the latter gives favorable fibrin as has been defined in the literature to be more efficient. However, more data is required to demonstrate the efficiency of the Ecarin-derived 3-D fibrin for supporting cell survival and growth toward specialized tissue regeneration.

Response to reviewers' comments

We thank all the reviewers for their reviews and constructive comments. To address all their questions and suggestions, we have conducted several new experiments and revised the manuscript accordingly, with changes highlighted in red. Here we include a point-by-point reply (in black) to all reviewers' comments (in *blue italics*).

Reviewer #1 (Remarks to the Author):

This paper reports a method for precise control of mechanical properties and porosity of fibrin-based scaffolds by using the snake venom-derived protein, ecarin, and prothrombin. The mechanical and structural properties were characterized by rheometer and the combined analysis of SANS and USANS. Moreover, by considering the endogenous activity as well as concentration of FXIII in fibrinogen purified from human plasma, the authors established the procedure to standardize batch-to-batch variations of the naturally derived enzyme. By selecting appropriate reaction conditions, the mechanical properties and porosity can be independently manipulated, from which they found the dominant role of fibrin architectures in the dictation of fibroblasts survival and differentiation. This work addressed an important issue about biophysical regulation of fibroblasts fates in 3D natural hydrogels and it was clearly demonstrated by their excellent molecular designs and characterization. This reviewer thinks the manuscript is suitable for publication in Nature Communications after addressing to the following minor concerns.

Reply: We thank the reviewer for the constructive suggestions on fibronectin quantification and fibrin permeability characterization. We have addressed all of the reviewer's questions and have supplemented the details as follows.

1. Even though the authors supplemented fibronectin to control biochemical cues, the activity of fibronectin is not only dependent on the concentration but also on the conformation as well as the degree of fibril formation. The authors need to discuss this point ideally by using several site-specific monoclonal fibronectin antibodies to exclude possible alteration of biochemical cues in their matrix.

Reply: We thank the reviewer for bringing up a key point that the precise localization of the fibronectin is not fully quantified. Although we did measure the total fibronectin concentration in our fibrin networks, we did not characterize if the added fibronectin is indeed incorporated into the 3D fibrin network. Following the reviewer's suggestion, we used one site-specific fibronectin monoclonal antibody (with appropriate controls to show specificity) to detect and localize fibronectin after fibrin formation by immunostaining. We find that fibronectin staining is co-localized with fibrin fibers. This suggests that fibronectin fibrils are indeed incorporated in the fibrin fibers and provide biophysical cues for cell adhesion and growth.

This result is similar to the findings of another study (<https://doi.org/10.1016/j.procbio.2018.09.013>) in which a similar pure fibrinogen plus fibronectin system was employed with fibronectin detected to be wrapped (or incorporated) to the fibrin matrix. We have added this new result in the main text (Line 240–247), Supplementary Figure S7b–d, and updated the methods accordingly (Line 20–21, 108–112, 123–127, 223–230).

2. The change in the porosity affects the diffusion of nutrients and waste products. Cellular traction forces exerted to the hydrogel matrices might further accelerate their diffusion. If this is the case, it is reasonable to see the effect of blebbistatin. To exclude this possibility, the authors need to evaluate the diffusion of some fluorescent probes, such as labeled dextran, through their scaffolds in the presence and absence of cells.

Reply: To address the reviewer's suggestion on detecting how the porosity of our fibrin system influences fibrin permeability, we have carried out further studies to understand how fibrin pore size affects fibrin permeability. We agree that it is important to exclude the possibility that the enhanced fibroblast differentiation and survival in the fine network S6 found in this study is due to the architecture-affected difference of diffusion of nutrients and wastes. Based on the reviewer's suggestion, we determined the permeability coefficient of both the S4 coarse and S6 fine networks. In addition, we detected the diffusion of dextran (10k and 70k MW) through these same fibrin networks (S4 and S6) with or without hydrostatic pressure. We find that under hydrostatic pressure (785 Pa, equivalent to ~8 cm of water depth), dextran completely permeabilized through the fibrin gels within 30–40 min. Without hydrostatic pressure, we found that after ~4 hours, ~30% of dextran (both 10k and 70k MW) has diffused through the fibrin, and by 1 day, the theoretical maximum diffusion amounts. These results suggest that though the permeability capacity of S4 is higher than S6 networks, mainly because of a higher flow rate for the S4 network, both S4 and S6 fibrin networks can efficiently diffuse the nutrients and waste products within the early time frame of cell spreading (the first day after 3D culturing).

Considering this highly efficient diffusion capacity and no difference in the diffusion capacity of the fibrin networks with different pore sizes, we did not further study how cellular traction force changes fibrin permeability/diffusion as cell traction-induced diffusion change might be subtle compared to the diffusion efficiency. We do agree with the reviewer's comment regarding cell traction force, that it is quite interesting and valuable, but it would require much more cell work and new models which are not closely related to the topic of the study.

The new results regarding the fibrin permeability/diffusion have been added to the main text (Line 266–284), Supplementary Figure S9, and Methods (Line 27–28, 38–39, 323–355).

Reviewer #2 (Remarks to the Author):

The manuscript entitled "Snake venom-defined fibrin architecture dictates fibroblast survival and differentiation" by Z. Wang et al. presents a study on natural fibrin hydrogels, which can be tuned in their architectural and mechanical properties so to produce an efficient and rapid wound sealant. These results can be significant as medical applications are likely and will provide a new way to reduce scar formation. However, I am surprised that another article of the group was published in June 2022 in Advanced Health Care Materials with clear relations with this manuscript, but not cited in this one (DOI: 10.1002/adhm.202200574). As the present study is anyway rather specialised, eventually such a journal might be better suited.

Reply: We appreciate that the reviewer finds this study has translational significance in wound healing and regenerative medicine and that ecarin has an advantage in controlling the formation of defined fibrin hydrogels compared to the current commercial thrombin-mediated fibrin products.

As the reviewer has mentioned, the application of ecarin in wound healing dressings was recently published (DOI: 10.1002/adhm.202200574) by our group. This published work reports the development of a rapid and potent bleeding control agent based on a polymer-ecarin system and investigated the first stage of wound healing, blood clot formation. The described work is very distinct from the present submission. In line with the reviewer's suggestion, we have added this relevant citation.

This manuscript describes the architecture and function of a defined fibrin matrix system that can be applied broadly to interrogate how biophysical parameters modulate cellular responses. We demonstrate the power of this matrix system on fibroblast behavior, function, and fate. We feel the work has a very broad and profound significance for numerous areas of tissue engineering, wound healing, and regenerative medicine, as well as basic mechanistic studies, to understand the underlying drivers of biophysical control over cellular responses. We strongly think a more comprehensive journal with a broader readership, like Nature Communications would be more suitable.

The authors have undertaken a series of complementary measurements to evidence the best network which produced the most promising results. The results are sound and statistical analysis was done with care. So I do not think that additional investigations will modify the conclusions. The methods used for the study are standard measurements, data analysis codes, they are correctly cited and well described. Due to space limitation, the Mat. And Meth. Section is a bit short, but it should be sufficient to allow to reproduce the data.

My main criticism concerns the abstract and introduction and I highly recommend that they should be reformulated. The abstract gives already technical details instead of resuming the aim of the study and the main results and possible applications. The introduction contains a lot of repetitions (the word "differentiation" appears 6 times in a short paragraph), although the scientific background permits a broad overview, which should be formulated in a more precise and attracting way.

Reply: As suggested by the reviewer, we have rewritten the abstract with an emphasis on the aim of the study and the main results and applications. For the introduction part, we have reduced word repetitions and redundancies. The specific changes are highlighted in red.

Minor points:

1) Figure 1b is mentioned after Figure 1f, only. The order should be adapted. I find the sub-figures of figure 1 too small.

Reply: Thanks for the feedback, accordingly we have reordered the sub-figures in Figure 1 and enlarged Figure 1 to make it clear

2) In line 103, 9 conditions corresponding to samples S1 – S9 are evoked, but in line 167 a sample S10 appears, which is not introduced.

Reply: We have added an annotation of sample S10 in Line 169.

3) A figure 4 appears two times, but figure 5 is missing.

Reply: We apologize for this error. This has now been corrected, and all the figure numbers are now corrected in the revised manuscript.

Reviewer #3 (Remarks to the Author):

The authors have conducted an extensive study highlighting the importance of fibrin stiffness and porosity for its performance as a regenerative scaffold. The importance of fibrin architecture and stiffness in regulating cell growth and differentiation. It has been well-established that a quick gelation makes softer hydrogel, which is tightly packed and not producing much porosity which is essential for cell permeation and growth. Essentially, the study's novelty is that the ecarin-induced gelation provides stiffness, mechanical properties, and porosity of the 3D architecture of the fibrin scaffold. In every study/evaluation the authors have compared the fibrin properties upon gelation with the physiological enzyme thrombin and the snake venom-derived Ecarin. However, the authors have used fibroblast as the model cell to evaluate and demonstrate that Ecarin controls the gelation time.

Reply: We thank the reviewer for the insightful comments and constructive suggestions on improving this study, in particular in investigating the role of focal adhesion-related questions. We have now specifically addressed each of the concerns made by the reviewer outlined below.

Overall, the study is promising and shows that venom-derived Ecarin is suitable for producing 3-D fibrin architecture for tissue regeneration. However, I have the following specific comments:

1. Generally, growth and differentiation of fibroblast is much easier and not much demanding as compared to other more specialized cells like keratinocytes, neural cells, or cardiomyocytes. Therefore, the use of fibroblasts in this study does not provide much value addition. The use of any one specialized cell as a model to demonstrate the cytocompatibility of Ecarin-induced fibrin could add more value.

Reply: We agree that demonstrating the compatibility of this system more broadly to potentially more sensitive cells is of great value. We employed primary fibroblasts in this study as they were relevant to the biological question of how the biophysical properties of fibrin influence wound healing outcome, specifically fibroblast-to-myofibroblast differentiation that dominantly contributes to scar formation or fibrosis during wound healing. Finding a biophysical way to control the fibroblast cell fate has significant promise in reducing scar formation or treating fibrosis. But as the reviewer is alluding to, this system has much broader applications.

As such, we have evaluated the cytocompatibility of ecarin-initiated fibrin on other specialized cells, including mesenchymal stem cells, endothelial cells, and dermal keratinocytes. We found that ecarin and/or textilinin have no evident effect on cell viability/proliferation after 7 days. These results have been added to the main text (Line 352–357), Supplementary Figure S13, and Methods (Line 30, 323–327). In addition, the *in vivo* systemic response evaluated in a different study (DOI: 10.1002/adhm.202200574) was also referenced, revealing short-term systemic immunocompatibility.

2. Another problem is the authors have missed the measurement of the fiber diameter which is important for the spreading of cells. The authors should analyze the focal adhesion using specific markers to demonstrate that the cell spreading is optimal.

Reply: We fully agree with the reviewer that the fiber diameter within the matrix will influence cell spreading and cell attachment to the fibrin network and the formation of focal adhesions. The specific evaluation of fiber radius (half of the fiber diameter), the data in Figure 1e, shows that the fiber radius/diameter is changing along with the pore size of the formed fibrin networks and is described together as 'architecture'. A coarse network (e.g., S1 network) has a larger average pore size and thicker fiber diameter, while a fine network (e.g., S6 network) has the opposite trend, as shown in Figure 1d–e (main text Line 107–115).

As suggested by the reviewer, we have now analyzed focal adhesion formation in different fibrin networks (S4 and S6). We first used immunostaining using focal adhesion-specific antibody vinculin and imaged 3D focal adhesions by confocal microscopy under high magnification. We found that the focal adhesion formation was matrix architecture and stiffness-dependent. We also monitored changes in vinculin expression by western blot and quantitated these changes. Supporting that focal adhesion-mediated cell spreading in a stiff and fine fibrin network is optimal compared to other conditions.

We have added these new results in the main text (Line 426–433), Supplementary Figure S18, and the Methods (Line 6–9, 21–24, 28–29, 281–298).

3. The confocal microscopy at higher magnification after staining for cytoskeletal actin could give information about the cell spreading to demonstrate if the stiffer fibrin formed using Ecarin supports focal adhesion.

Reply: We agree that higher magnification will provide greater clarity to these figures. This has now been provided in the new Figure S18, as part of our last reply. The reviewer's point is even more important from the mechanobiology perspective and gives one potential explanation of architecture-regulated fibroblast differentiation. Fibrin pore size and stiffness affect fibroblast responses by influencing focal adhesion formation, which dictates mechano-sensing and mechano-transduction. We have addressed this in the main text (Line 426–433), Supplementary Figure S18, and the Methods (Line 230–233, 259–261).

4. In the confocal microscope, the adhered fibroblast are spatially far apart and there is no evidence if the cells can grow and cover the porous structures to form functional tissue with deposition of ECM proteins such as collagen and elastin.

Reply: In this study, we use a relatively low cell density ($0.5\text{--}1\times 10^5\text{ mL}^{-1}$) and grow cells for a relatively short time (7 days) to avoid cell–cell contact and overgrowing into clusters. Therefore, it would be difficult for fibroblasts to grow and form functional tissue. The reason is that we aim to isolate matrix architecture and stiffness cues to study fibroblast survival and differentiation specifically. Cell–cell contact, or cluster formation because of cell overgrowing, would introduce numerous extra cues and further complicate the study.

We agree that matrix production of fibroblasts has vital functions in wound healing. Therefore, as suggested by the reviewer, we detected collagen production in fibrin networks by immunostaining, as collagen is the major matrix produced by fibroblasts and is closely related to wound healing outcomes. We find that cells produce more collagen in the stiff fine fibrin S6 compared to other conditions. Collagen was observed to localize not only in the cytoplasm but

also deposited in the fibrin matrix. Therefore, the functional deposition of fibroblasts in the fibrin matrix is dependent on matrix architecture and stiffness.

These results have been added to the main text (Line 381–384), Supplementary Figure S15, and Methods (Line 24–25, 233–238, 261–262).

5. The authors have not shown if the cells adhere to the superficial fibrin only, or if it has migrated to the internal strands of the 3-D fibrin architecture. The authors could analyze the 3-D structure using a z-sectioning feature of confocal microscopy.

Reply: We understand that using the word 'encapsulation' when we describe cell 3D growing in the fibrin networks may come across as confusing and is usually used in the 2D cell culture model. In this study, we use the 3D cell culture model, which means fibroblasts are growing in the 3D fibrin architecture after the fibrin is formed. Therefore, the word 'encapsulation' is replaced by '3D cell growth' in the main text and Methods section.

6. The authors have systematically collected a lot of data on the fibrin gelation time, fiber stiffness, and that the mechanical properties are improved by replacing thrombin with Ecarin. The comparisons with thrombin and prothrombin etc. (with and without FXIII) with the 3-D fibrin formed using Ecarin, clearly present that the latter gives favorable fibrin as has been defined in the literature to be more efficient. However, more data is required to demonstrate the efficiency of the Ecarin-derived 3-D fibrin for supporting cell survival and growth toward specialized tissue regeneration.

Reply: We thank the reviewer's comment concerning the tissue regeneration capacity of fibroblasts in our fibrin system. We think this question can be answered by Comment 4. The collagen expression in the ecarin-initiated 3D fibrin is confirmed by confocal microscopy, though the cell proliferation curve does not show evidence of cell growth because of the low cell density and short growth time.

REVIEWERS' COMMENTS

Reviewer #3 (Remarks to the Author):

The authors have addressed all review comments appropriately with additional data, wherever required.

The redressed manuscript is well-presented.

I have no more concerns